# TEST-TIME SCALING IN DIFFUSION LLMS VIA HIDDEN SEMI-AUTOREGRESSIVE EXPERTS

**Jihoon Lee**[1], **Hoyeon Moon**[1], **Kevin Zhai**[5], **Arun Kumar Chithanar**, **Anit Kumar Sahu**[2], **Soummya Kar**[3], **Chul Lee**, **Souradip Chakraborty**[4][‡], **Amrit Singh Bedi**[5][‡]

[1]Yonsei University, [2]Oracle, [3]Carnegie Mellon University,
[4]University of Maryland, [5]University of Central Florida

## ABSTRACT

Diffusion-based large language models (dLLMs) are trained to model extreme flexibility/dependence in the data-distribution; however, how to best utilize this at inference time remains an open problem. In this work, we uncover an interesting property of these models: dLLMs trained on textual data implicitly learn a mixture of semi-autoregressive experts, where different generation orders reveal different specialized behaviors. We show that committing to any single, fixed inference time schedule, a common practice, collapses performance by failing to leverage this latent ensemble. To address this, we introduce HEX (Hidden semi-autoregressive EXperts for test-time scaling), a training-free inference method that ensembles across heterogeneous block schedules. By doing a majority vote over diverse block-sized generation paths, HEX robustly avoids failure modes associated with any single fixed schedule. On reasoning benchmarks such as GSM8K, it boosts accuracy by up to 3.56× (from 24.72% to 88.10%), outperforming top-K margin inference and specialized fine-tuned methods like GRPO, without additional training. HEX even yields significant gains on MATH benchmark from 16.40% to 40.00%, scientific reasoning on ARC-C from 54.18% to 87.80%, and TruthfulQA from 28.36% to 57.46%. Our results establish test-time scaling as a powerful principle for dLLMs, showing that the sequence in which masking is done can play a significant role in test-time scaling/inferencing of dLLMs.[1]

## 1 INTRODUCTION

Diffusion-based large language models (dLLMs) are rapidly emerging as a promising alternative to traditional autoregressive LLMs generalizing beyond the next token prediction (Nie et al., 2025b). Unlike autoregressive models, dLLMs generate text via an iterative mask-and-unmask process, allowing them to decode tokens in essentially arbitrary order (Kim et al., 2025). This fundamental change in the generation mechanism during training grants dLLMs remarkable flexibility at inference time. In fact, recent dLLMs have already demonstrated competitive (and sometimes superior) performance compared to their autoregressive counterparts on a similar scale (Zhao et al., 2025). These early successes indicate that the masking strategy during inference plays a crucial role.

**Gaps in our understanding about dLLMs.** The freedom to choose the generation order, *the masking strategy*, is the central advantage of dLLMs. Recent works (Kim et al., 2025; Nie et al., 2025b) have tried to harness this flexibility by relying on prediction confidence, progressively unmasking high-confidence tokens (top-K margin in Figure 1). However, such an approach often leads to inherently biased solutions, as it overlooks the crucial sequential structure in the language training data, thereby inducing implicit biases in the learned masking strategies. As a result, these methods might perform worse than random unmasking, as shown in Figure 1. Why this happens and what we can do to avoid such a failure remains an open question, which we address in this work.

**Our key finding: hidden semi-autoregressive experts.** We uncover a new dimension of test-time scaling for diffusion LLMs, centered on the masking strategy. We observe that the central degree

---

[‡]Joint supervision.
[1]The source code is available at `https://github.com/junos-ai-org/Test-Time-Scaling`.

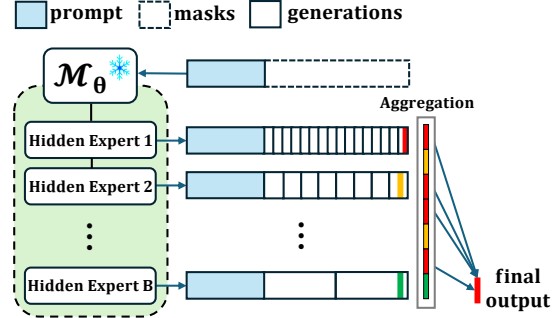 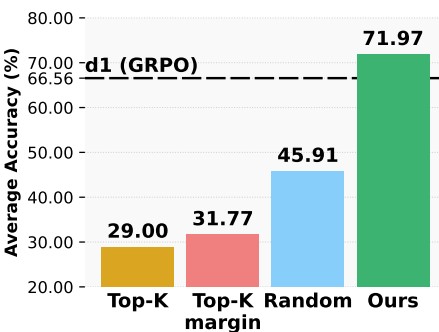

Figure 1: Overview of our proposed HEX framework. **Left**: HEX leverages multiple semi-autoregressive hidden experts, guided by different masking schedules, to produce concatenated outputs and a final answer. **Right**: HEX outperforms Top-K, Top-K margin (Kim et al., 2025) and Random expert selection strategies (Nie et al., 2025b) on reasoning tasks (GSM8K, MATH, ARC-C), surpassing the training-based GRPO baseline (d1) (Zhao et al., 2025).

of freedom during inference lies in choosing the *masking strategy*, which is unique to dLLMs and critically shapes the generation distribution. The sequential nature of language data causes dLLMs to implicitly learn a mixture of semi-autoregressive experts during training. Each of these "experts" is naturally biased towards distinct masking distributions at test time, with a natural preference toward semi-autoregressive generation. For the first time, we demonstrate that this latent mixture can be deliberately accessed during inference. By varying the block size used in semi-autoregressive decoding, we can activate different experts, mirroring the conditions the model saw during training. This insight unlocks a novel method for test-time scaling. By marginalizing across these block schedules, we can exploit the latent ensemble of experts, resulting in significantly more robust and optimal inference.

Hence, we propose HEX (Hidden semi-autoregressive EXperts), a training-free inference method that uncovers a new dimension of test-time scaling for dLLMs. HEX marginalizes across block schedules, treating block size and order as latent variables that define an additional scaling dimension, and aggregates predictions via majority voting. In doing so, it robustly avoids the pitfalls of committing to any single decoding path, turning dLLMs' hidden flexibility into a principled mechanism for test-time scaling. We summarize our contributions as follows.

**(i) New dimension of test-time scaling in dLLMs.** We show that dLLMs implicitly learn a mixture of semi-autoregressive experts, and that block scheduling helps to uncover this latent structure. (Section 3).

**(ii) HEX: Hidden semi-autoregressive EXperts for test-time scaling.** We introduce HEX, a training-free inference algorithm ensembling over semi-autoregressive schedules with majority-vote aggregation (Algorithm 2), turning ordering into a reliable test-time scaling dimension. (Section 4).

**(iii) Comprehensive experimental analysis: matching GRPO-level performance.** HEX achieves GRPO-level results on GSM8K, MATH, ARC-C, and TruthfulQA, without retraining, establishing test-time scaling as a powerful new paradigm for diffusion LLMs. HEX outperforms existing state-of-the-art inference methods (Kim et al., 2025) on reasoning tasks, boosting accuracy by up to 3.56× (from 24.72% to 88.10%.) HEX even produces massive gains on more challenging tasks, including MATH (Lightman et al., 2023) (from 16.40% to 40.00%), ARC-C (Clark et al., 2018) (from 54.18% to 87.80%), and TruthfulQA (Lin et al., 2021) (from 28.36% to 57.46%). (Section 5).

## 1.1 RELATED WORK

**Diffusion Large Language Models.** Diffusion models have achieved state-of-the-art performance in image generation (Ho et al., 2020; Song et al., 2020), and recent advances extend them to the discrete domain of language. The early approaches applied continuous diffusion to latent text representations (Austin et al., 2021; Li et al., 2022; Dieleman et al., 2022), but faced challenges with scalability and discretization. A masked diffusion paradigm soon emerged as a more tractable dis-

crete alternative (Nie et al., 2025a), with large-scale implementations such as DiffuLLaMA (Gong et al., 2024) and LLaDA (Nie et al., 2025b) demonstrating that diffusion LLMs (dLLMs) can rival similarly sized autoregressive models, even on complex reasoning (Zhao et al., 2025; Tang et al., 2025). This potential extends even to multimodal understanding (You et al., 2025; Wen et al., 2025).

**Inference-Time Methods for dLLMs.** In autoregressive models, inference-time scaling has been extensively studied, ranging from chain-of-thought prompting (Wei et al., 2022) and self-consistency (Wang et al., 2022) to scaling the allocation of test-time compute (Snell et al., 2024). In contrast, inference-time methods for dLLMs remain sparse. Most gains in dLLM performance so far have come from training-time improvements, such as applying GRPO (Zhao et al., 2025; Tang et al., 2025) or other post-training methods (Wang et al., 2025; Asano et al., 2026; Zhai et al., 2026).

## 2 PROBLEM FORMULATION

**Masked Diffusion Language Models (MDM).** Let $x = (x_1, \ldots, x_n) \in \mathcal{V}^n$ be a length-$n$ token sequence over vocabulary $\mathcal{V}$. A masked diffusion large language model (dLLM) specifies a conditional denoiser $p_\theta(x[M] \mid x[M^c])$ for any mask $M \subseteq [n]$, where $M^c = [n] \setminus M$. The notation $x[M]$ is defined as the sequence of tokens from $x$ on the indices of $M$, i.e. $x[M] = (x_i)_{i \in M}$. In the forward (corruption) process, a random subset of tokens $M \subseteq [n]$ is masked (replaced by a special symbol [MASK], and model $p_\theta$ is tasked with recovering the original tokens in $M$ given the unmasked tokens in the complement $M^c := [n] \setminus M$. Formally, for a random mask pattern $M$, the model produces a conditional distribution $p_\theta(x[M] \mid x[M^c])$ on the masked tokens. Here, $x[M]$ denotes the masked tokens in $x$. The training objective is to maximize the likelihood of ground-truth tokens in these masked positions. Hence, the training problem can be written as

$$\theta^* \in \arg\min_\theta \mathcal{L}_{\text{mask}}(\theta) := \mathbb{E}_{x \sim \mathcal{D}} \mathbb{E}_{\ell \sim \text{Unif}([n])} \mathbb{E}_{M \subseteq [n], |M| = \ell} \left[ -\sum_{i \in M} \log p_\theta(x_i \mid x[M^c]) \right]$$

$$= \left[ -\sum_{M \subseteq [n], \, i \in M} \frac{1}{|M|} \frac{1}{\binom{n}{|M|}} \mathbb{E}_{x \sim \mathcal{D}} \left[ \log p_\theta(x_i \mid x[M^c]) \right] \right], \tag{1}$$

where $\mathcal{D}$ is the data distribution, $\ell \sim \text{Unif}([n])$ is a uniformly sampled number of masked tokens $\ell \in \{1, \ldots, n\}$ and $M \subseteq [n]$ is the randomly selected subset of length $|M| = \ell$. The summation in (1) runs over all masked token positions $i \in M$, and the loss on each such position is the negative log-likelihood of the true token $x_i$ given the remaining context (unmasked) $x[M^c]$. The objective in (1) trains the model to predict randomly masked-out tokens, and can be viewed as averaging next-token losses over all token permutations, i.e., an any-order objective (Nie et al., 2025b).

**Inference as the Core Challenge.** After learning $\theta^*$ from (1), generation happens step by step. For instance, for a given prompt $x_{\text{prompt}}$, it starts by selecting the number of tokens to be generated (say $L$), and then requires choosing *how* to reveal tokens. Let a *decoding trajectory* be a sequence of masks $\tau = (M_1, \ldots, M_T)$ that partition $[n]$ (such that $\bigsqcup_{t=1}^T M_t^c = [n]$), with per-step sizes $\ell_t = |M_t|$. At step $t$, the model predicts all

---

**Algorithm 1** Vanilla MDM Inference

**Input:** prompt $x_{\text{prompt}}$, output length $L$, steps $T$; mask schedule $\{M_t\}_{t=1}^T$, model $p_\theta(\cdot | \cdot)$.

**Initialize:** $x^{(0)} \leftarrow [\text{MASK}]^{\times L}$;
  **for** $t = 1, 2, \ldots, T$ **do**
    Predict all masked tokens simultaneously via $\sim p_\theta\left(\cdot \mid [x_{\text{prompt}}, x^{(t-1)}]\right)$
    $x^{(t)} \leftarrow$ Fill with predicted tokens
    Fix tokens at location $i \in M_t^c$
    Mask tokens at location $i \in M_t \setminus \left(\cup_{k=1}^{t-1} M_k^c\right)$

**Output:** $x^T$

---

masked tokens conditioned on the currently revealed context $x\left[\cup_{s=1}^{t-1} M_s^c\right]$. For a fixed trajectory $\tau$ and prompt $x_{\text{prompt}}$, a functional of natural log-likelihood is

$$\mathcal{J}(\tau; \theta \mid x_{\text{prompt}}) = \sum_{t=1}^T \sum_{i \in M_t} \log p_{\theta^*}\left( x_i \mid x_{\text{prompt}}, x\left[ \bigcup_{s=1}^{t-1} M_s^c \right] \right). \tag{2}$$

We summarize the vanilla inference procedure in Algorithm 1. The ideal (but empirically intractable[2]) goal is to choose $\tau$ such that we obtain a sample which maximizes $\mathcal{J}(\tau; \theta \mid x_{\text{prompt}})$.

Because (1) trains on *all* mask patterns, many training subproblems are intrinsically ill-posed (e.g., extremely large masks with scant context) due to the implicit sequential bias in the language training distribution. For example, some conditionals are rarely observed or provide little meaningful context, making them effectively unsolvable. As a consequence, the model ends up learning only a subset of subproblems or conditionals, while others remain poorly learned or ignored. A uniform masking objective forces the model to put equal weight on every subproblem, including those that confound the masking sequence, resulting in a suboptimal masking strategy overall. This mismatch creates a gap at inference time: the model's behavior becomes highly sensitive to the masking schedule, and strong performance depends on selecting strategies that align with the sequential biases implicitly learned during training.

**Key Open Question.** Thus, the central question becomes: *how can we design an inference strategy that faithfully reflects what the model has learned during training, given that dLLMs leave the process fundamentally under-specified and require us to decide the optimal masking trajectory.*

## 3 LIMITATIONS OF SOTA AND OUR KEY INSIGHT

**Failure of existing inference methods for reasoning tasks.** The existing dLLMs rely on heuristic inference-time strategies to choose which tokens to unmask at each denoising step. The common approaches include random sampling (randomly picking masked positions to predict) and confidence-based selection (choosing the token positions with the high model confidence or probability). For example, Kim et al. (2025) showed that for Sudoku puzzles a simple confidence-based top-K margin method can boost accuracy from 7% to nearly 90%. Intuitively, one might expect a similar benefit for reasoning tasks. Surprisingly, we find the opposite in reasoning benchmarks. In our experiments (see Figure 2) on GSM8K math problems, random unmasking far outperforms top-K margin (high-confidence) decoding. Instead of guiding generation, high confidence derails the unmasking trajectory into producing degenerate outputs. In Figure 2, the top-K margin strategy consistently unmasked the [AfterEoT] ($\langle\text{endoftext}\rangle$) token at all positions, proceeding *backward from the end to the front* (See A.2 for a detailed explanation).

This leads to degenerate outputs (shown in red text) in Figure 2. This surprising result challenges our intuition built through studying prior work.

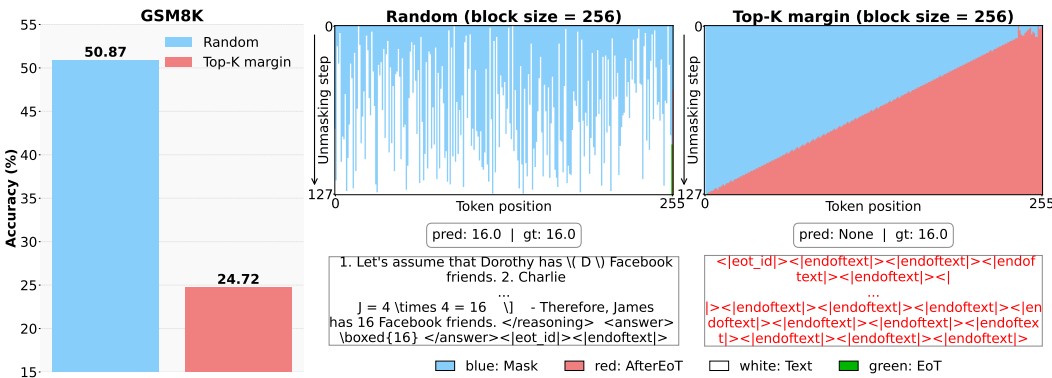

Figure 2: **Random vs. Top-K margin inference on GSM8K. Left**: Random decoding achieves 50.87% accuracy, while **Right**: Top-K margin only 24.72%. For each method, the text box shows the result at the last unmasking step. Top-K margin generates output tokens in reverse, from the end toward the beginning, and exhibits a catastrophic collapse in which all tokens are [AfterEoT] (shown in red). Over 55.5% of top-K margin runs suffered this collapse, yielding very low accuracy. These failures cast doubt on methods that rely solely on token confidence.

---

[2]Because simply making consecutive greedy choices does not guarantee that the overall $\mathcal{J}$ is maximized, and performing a global search would require evaluating all possible trajectories, which is practically infeasible.

**Unexpected reversal of intuition.** Our findings highlight a key limitation of relying on common heuristics from autoregressive models and prior work (Kim et al., 2025). While one would expect that "follow the model's own highest-probability tokens" is a reliable strategy, our results show that methods relying solely on token confidence are not sufficient for strong performance in complex reasoning tasks. Our results thus raise basic questions: Why does random sampling outperform top-K margin? Why does the model overconfidently pick the [AfterEoT] so early? To answer these questions, we propose a new perspective on the dLLM's internal structure. Our key insight is that the dLLM can be viewed as an implicit mixture of experts, which allows us to mitigate the risk of overconfidently predicting [AfterEoT] tokens too early. By aggregating predictions from experts conditioned on different subsets of tokens, effectively marginalizing over contexts, our approach avoids prematurely committing to high-confidence tokens like the [AfterEoT] token.

**From failure to mechanism.** The surprising failure of confidence-based schedules suggests that local token probabilities are unreliable under many masked contexts induced at inference. Because the dLLM objective (in (1)) averages over a wide variety of maskings, including ill-posed ones, some conditionals are poorly learned and disproportionately favor special tokens such as [AfterEoT]. Our view is to treat each semi-AR block schedule as querying a different conditional expert, then marginalize across experts to recover robust predictions. This replaces brittle confidence-following with consensus-seeking and is the core rationale behind HEX.

**Our Key Insight: dLLM is an implicit mixture of experts.** From (2), it is evident that the diffusion LLM training leads to a model with a family of masked–token conditionals

$$\big\{\, p_\theta(x_i \mid [x_{\text{prompt}}, x[U]]) : i \in [n],\ U \subseteq [n] \setminus \{i\} \,\big\},$$

which we can view as implicit "experts" indexed by the visible/unmasked set of tokens $U$. For a fixed target index $i$ at test-time, the ideal goal of inference is to aggregate the relevant experts $\{p_\theta(x_i \mid [x_{\text{prompt}}, x[U]])\}_{U \subseteq [n] \setminus \{i\}}$ into a single prediction rule. A principled aggregation strategy is the mixture-of-experts predictor given by

$$p_{\text{mix}}(x_i = a \mid x_{\text{prompt}}) \;=\; \sum_U \pi(U \mid x_{\text{prompt}})\, p_\theta(x_i = a \mid [x_{\text{prompt}}, x[U]]), \tag{3}$$

where $\pi(U \mid x_{\text{prompt}})$ denotes the weight or trust placed on expert $U$ for prompt $x_{\text{prompt}}$.

**A Toy Example.** In our toy example (Figure 3), we examine the model's answer to the prompt "Who invented telephone?" (ground truth: "The inventor was Bell."). We treat each latent conditional context (or expert) as a separate setting that produces a distribution to predict masked tokens. Figure 3 plots these distributions for a particular position, $i = 4$ (the token 'Bell'). As the plot makes clear, most experts concentrate probability on "Bell" (the correct token), but a few contexts produce flat or incorrect distributions that never predict "Bell".

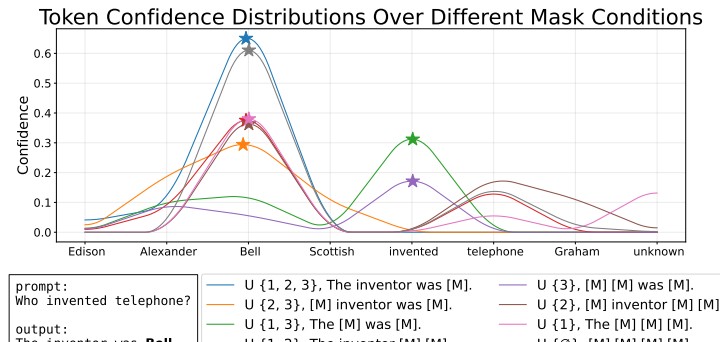

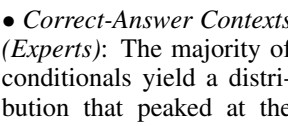

Figure 3: The distribution of the 4th token 'Bell' in the output sequence changes significantly depending on the $2^3$ masking conditions applied to the previous three tokens: 'The', 'inventor', 'was'. The star mark indicates the highest confidence for each distribution generated by $U$. Some masking conditions (violet and green) produce collapsed distribution: "Bell Bell was invented." (ungrammatical sentence), "The telephone was invented." (missing target information), respectively.

• *Correct-Answer Contexts (Experts)*: The majority of conditionals yield a distribution that peaked at the correct answer token Bell. These contexts effectively act as "experts" on this question.

• *Non-Expert Contexts*: Some conditionals do not produce a clear peak in 'Bell'. These contexts fail to predict the correct answer.

According to our insight (3), dLLMs learn to unmask uniformly random masked patterns during training. As a result of this randomized training, the accuracy varies across conditional contexts (or experts). Ideally, we would like to construct $p_{\text{mix}}$ that maximizes accuracy by accounting for the importance of each expert. However, directly estimating $p_{\text{mix}}$ is challenging. This is because doing so would require likelihood estimation over all ($2^{N-1}$ for the $N$-th token) conditional contexts (or experts), and the hidden gating distribution $\pi(U)$ that governs how likely each expert yields the correct answer is unknown and not directly observable. Meanwhile, the toy example (Figure 3) intuitively suggests that it may be possible to directly pursue the final objective of selecting the most accurate token without explicitly constructing $p_{\text{mix}}$. When each masking condition is regarded as an individual intrinsic expert, Figure 3 shows that *majority* of experts "know" the answer.

## 4 HIDDEN SEMI-AUTOREGRESSIVE EXPERTS FOR TEST-TIME SCALING

In addition to the exponentially large number of experts required for estimation, prior work (Tang et al., 2025; Zhao et al., 2025; Nie et al., 2025b) has reported difficulties in likelihood estimation in dLLMs. Ideally, we would compute the Bayes-optimal conditional probability $p_{\text{mix}}(x_i = a \mid x_{\text{prompt}})$ by fully marginalizing over all possible sequences of tokens in $U$. To sidestep this, we approximate the ideal mixture by ensemble-averaging over a small set $B$ of "semi-autoregressive," each defined by a particular block-schedule $b \in B$. Concretely, we sample several semi-AR decoding schedules $b$, each of which queries exactly one expert $U_b$, and then averaging:

$$p_{\text{mix}}(x_i = a \mid x_{\text{prompt}}) \approx \mathbb{E}_{b \sim B}\big[p_\theta(x_i = a \mid [x_{\text{prompt}}, x[U_b]])\big]. \tag{4}$$

The final prediction can be made using the Bayes rule:

$$\hat{a} = \arg\max_a \; p_{\text{mix}}(x_i = a \mid x_{\text{prompt}}). \tag{5}$$

This simple Monte Carlo approximation can be further simplified via majority voting: draw one sample $\hat{a}_b$ from each queried expert $U_b$, and return the mode of the sampled values. Thus, either by mixture averaging or majority vote, test-time aggregation amplifies the correct prediction by combining the specific conditionals (experts). The effectiveness of this approximation is intuitively illustrated in Figure 4. Based on this insight, we summarize our proposed approach in Algorithm 2.

For each schedule, we first convert the final generated token sequence $x^{(T)}$ into a natural language string representation. We then apply numeric parsing to remove LaTeX formatting, whitespace, and commas. This yields a parsed output $f(x^{(T)})$ for each schedule, which we store in a list. The final answer is chosen as the value that occurs most frequently across schedules. If there is a tie (i.e., two or more values appear with the same highest frequency), we choose the value generated from the schedule with the smallest block size.

---

**Algorithm 2 H**idden semi-autoregressive **EX**perts for test-time scaling

**Input:** prompt $x_{\text{prompt}}$, output length $L$, output parser $f$, steps $T$, experts $B$; semi-AR mask schedule $\{\{U_{t,b}\}_{t=1}^T\}_{b=1}^B$, model $p_\theta(\cdot|\cdot)$.

Initialize `outputs` $\leftarrow [\,]$
**for** $b = 1, 2, \ldots, B$ **do**
    **Initialize:** $x^{(0)} \leftarrow [\text{MASK}]^{\times L}$
    **for** $t = 1, 2, \ldots, T$ **do**
        Predict all masked tokens simultaneously via $\sim p_\theta\big(\cdot \mid [x_{\text{prompt}}, x^{(t-1)}]\big)$
        $x^{(t)} \leftarrow$ Fill with predicted tokens
        Fix tokens at location $i \in U_{t,b}^c$
        Mask again tokens at location $i \in U_{t,b} \setminus \left(\cup_{k=1}^{t-1} U_{k,b}^c\right)$
    $x^{(T)} \leftarrow x^{(T)}$ as a natural language string
    Append parsed string output $f(x^{(T)})$ to `outputs`
**Output:** max(`outputs`, key = lambda x: (`outputs`.count(x), -`outputs`.index(x)))

---

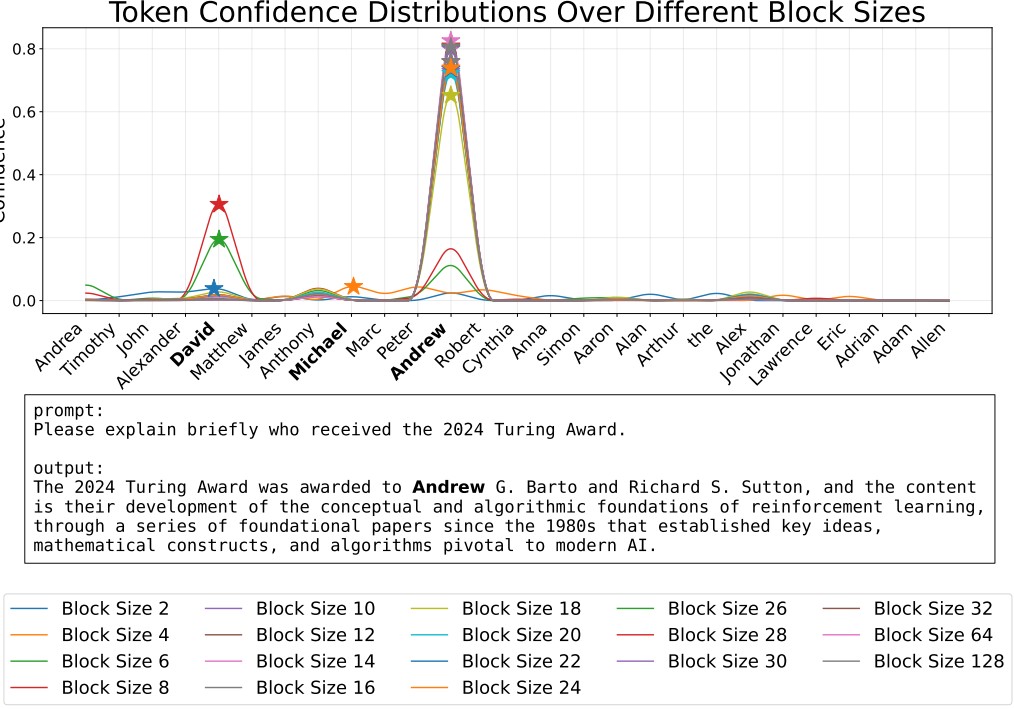

Figure 4: When asked about the 2024 Turing Award winners, names other than the actual recipients (such as Michael or David) might be generated due to different block sizes, which in turn risks producing incorrect information in the subsequent token sequence. However, if we generate outputs with various block sizes and then select the most frequently produced answer, that answer is more likely to be correct, since it was probably derived through a valid reasoning (Andrew) during the inference process.

**Why semi-autoregressive?** Diffusion LLMs allow all trajectories to reveal masked tokens, but uniformly random orders are suboptimal for language: they create unnatural partial contexts that the model was never intended to generate at test time. A practical restriction is semi-autoregressive left-to-right decoding (semi-AR): fix a block size $b \in \{1, \ldots, B_{\max}\}$ (where $B_{\max} = n$) and partition $[n]$ into consecutive blocks

$$M_t = \{(t-1)b+1, \ldots, \min(tb, n)\},$$

for $t = 1, \ldots, T(b) = \lceil n/b \rceil$, revealing blocks left-to-right while denoising within each block via diffusion. This preserves a strong prefix structure (natural for language), yet allows parallel denoising inside a block. Empirically, we find (see Table 1) that semi-AR decoding avoids pathologies seen in fully parallel decoding. In particular, when using a single large block (i.e., non-semi-AR parallel decode), we often observe an [AfterEoT] collapse: the model erroneously floods the tail with [AfterEoT] tokens or repeats (Figure 8). By contrast, constraining to moderate block sizes (decoding left-to-right) eliminates this collapse and dramatically improves accuracy. (See Table 1: semi-AR has 0% collapse and much higher accuracy than non-semi-AR decoding.) Intuitively, focusing first on the left part of the output prevents the model from prematurely committing to a length or drifting with high-confidence tail tokens.

Table 1: Semi-AR based decoding eliminates [AfterEoT] collapse and improves accuracy.

| Dataset | Collapsed ($\downarrow$, %) | Accuracy ($\uparrow$, %) |
|---|---|---|
| *Baseline (non–semi-AR)* | | |
| GSM8K | 55.80 | 22.52 |
| MATH | 29.80 | 16.60 |
| *Semi-AR* | | |
| GSM8K | **0.00** | **76.27** |
| MATH | **0.00** | **32.80** |

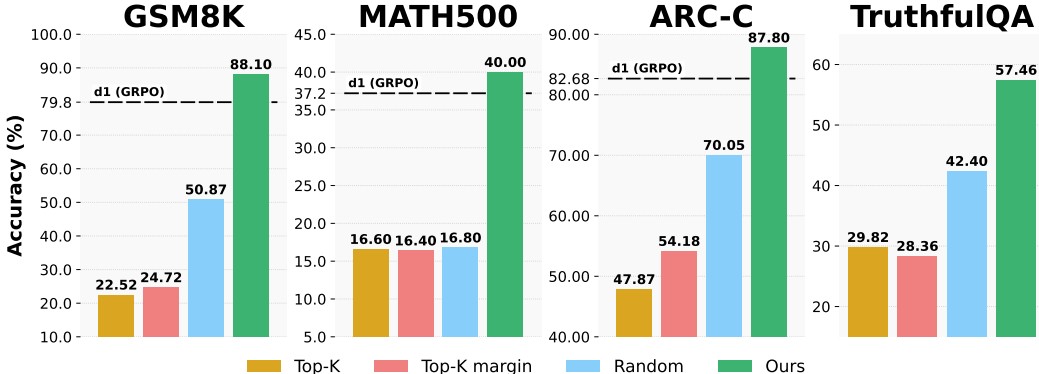

Figure 5: **HEX improves reasoning accuracy.** On LLaDA-8B-Instruct, HEX outperforms training-free baselines (Random, Top-$k$, Top-$k$-margin) on GSM8K, MATH, ARC-C, and TruthfulQA. In GSM8K, MATH, ARC-C, it even outperforms the model trained with GRPO without any training.

## 5 EXPERIMENTS

In this section, we empirically validate our claims. (i) *Effectiveness*: We first demonstrate that HEX significantly outperforms existing training-free and fine-tuned methods on a suite of reasoning benchmarks. (ii) *Scaling behavior:* We then analyze the performance-computation trade-off, showing how accuracy scales with more diverse generation paths. (iii) *Working mechanism:* Finally, through a series of ablations and qualitative examples, we explore the mechanisms behind HEX's success, confirming that its gain comes from ensembling a latent mixture of semi-AR experts rather than relying on heuristics like model confidence.

### 5.1 SETUP

**Datasets and Metrics.** We follow standard reasoning benchmarks: **GSM8K** (Cobbe et al., 2021) consisting of high-quality problems with diverse linguistic expressions, **MATH** (Lightman et al., 2023) is a more challenging math benchmark that includes competition-level math problems, **ARC-C** (Clark et al., 2018) is the Challenge Set from AI2's ARC dataset, consisting of science knowledge-based questions that are difficult to solve with simple keyword matching or retrieval, and **TruthfulQA** (Lin et al., 2021) which evaluates the tendency of language models to generate false information by following human misconceptions or false beliefs.[3] Primary metric is task accuracy.

**Models and Baselines.** All experiments with inference methods were performed using the LLaDA-8B-Instruct model (Nie et al., 2025b), and the application of d1 (GPRO) (Zhao et al., 2025) is subsequently based on this model. For all methods, when the output length is 256 tokens, the number of unmasking steps is 128. At each step, two masked tokens are unmasked, and this process is repeated until all tokens are revealed. *Random* unmasks two randomly chosen masked tokens per step. Top-$k$ margin unmasks, at each step, the two masked tokens with the highest margin defined as (top-1 confidence − top-2 confidence) at their positions. d1 (GRPO) row uses the reported best value (Zhao et al., 2025) for GSM8K and MATH, and for ARC-C we report a value reproduced after 1 epoch of training. TruthfulQA trained on d1 (GRPO) is excluded because there is no training data available, and neither were checkpoints released. HEX draws five samples at temperature = 0.9 for each of the block sizes $[8, 16, 32, 64, 128]$, yielding 25 samples in total. If a tie occurs for the most frequent value, the value generated with the smallest block size is selected (Algorithm 2).

### 5.2 MAIN RESULTS: HEX ESTABLISHES A NEW STATE-OF-THE-ART

**Overall performance.** Figure 5 shows that HEX achieves the strongest results across all four reasoning benchmarks, outperforming both training-free and fine-tuned baselines. Compared to existing decoding strategies (Nie et al., 2025b; Kim et al., 2025), HEX delivers large and consistent

---

[3]We use official evaluation scripts; numeric parsing strips LaTeX wrappers/whitespace/commas.

gains. In GSM8K, for example, HEX reaches 88.10% accuracy, far higher than Random decoding (50.87%) and Top-$k$ margin (24.72%). These results show that confidence-based heuristics are unreliable in diffusion LLMs, whereas consensus-based voting in HEX is robust (Figure 7).

**Comparison with GRPO fine-tuned models.** Perhaps most strikingly, HEX also surpasses d1 (GRPO), which requires costly reinforcement learning fine-tuning. On GSM8K (88.10% vs. 79.80%), MATH (40.00% vs. 37.20%), and ARC-C (87.80% vs. 82.68%), HEX sets a new state of the art without updating model parameters.

Intuitively, fixed inference scheduled in existing techniques sometimes asks the model to guess hard tokens too early, which leads to mistakes. In contrast, HEX tries several semi-autoregressive schedules and then picks the answer that many schedules agree on. In practice, answers that show up across schedules are more reliable than answers from any single schedule.

**Takeaway.** These results suggest that the reasoning ability of a diffusion LLM remains latent and can be unlocked at inference time through block-marginalized ensembling, without any fine-tuning.

## 5.3 ANALYSIS OF SCALING AND COMPUTE TRADE-OFF

Figure 6 shows that HEX's accuracy improves monotonically as the number of voting samples increases, while the tie rate, an indicator of ambiguity, steadily declines. Intuitively, different semi-AR schedules make different mistakes but tend to agree on the correct answer; adding schedules cancels schedule-specific errors and strengthens consensus, so ties resolve and accuracy improves. This trend holds consistently across all four benchmarks. Because sampling more trajectories linearly increases compute cost, HEX effectively exposes a tunable accuracy, compute knob: practitioners can trade inference cost for accuracy in a predictable way, without retraining.

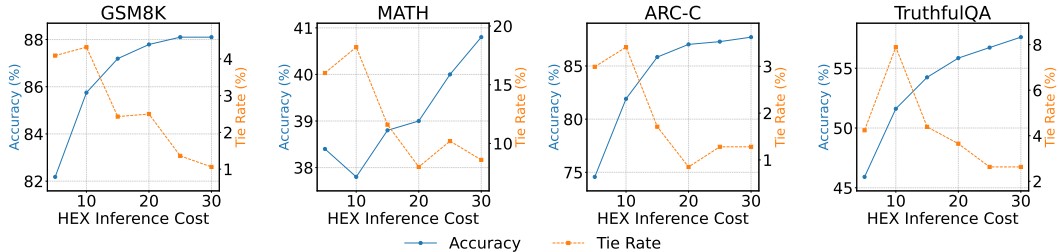

Figure 6: As the number of majority voting samples in HEX increases, accuracy improves and the tie rate decreases. The block sizes used are [8, 16, 32, 64, 128], and sampling was performed while increasing the number of seeds (1-6).

**Takeaway.** HEX not only establishes state-of-the-art performance but also provides a principled mechanism for test-time scaling, ensuring accuracy improves with more inference budget.

## 5.4 ABLATION STUDIES

Next, we analyze the mechanisms behind the HEX improvements, focusing on two key factors: the role of block diversity and the role of likelihood versus frequency in candidate selection.

**Effect of block diversity.** Beyond using a fixed set of block sizes, we test whether ensembling over more varied (and even randomly generated) block schedules further boosts performance. As shown in Table 2, increasing the number of dynamic trajectories from 5 to 30 on GSM8K improves accuracy from 81.96% to 84.15% while reducing the tie rate to less than half. This reinforces our hypothesis that performance gains come from aggregating diverse "semi-AR experts." We note that diversity matters, but structured diversity (fixed block set with multiple seeds) is even stronger (as in Table 3), yielding the highest overall gains.

**Frequency vs. likelihood.** We then examine whether HEX's gains could simply come from likelihood-based re-ranking.

Table 2: Accuracy and tie rate (%) on GSM8K across *dynamic* block sizes. See Figure 12 for details on block sizing.

| Size | Accuracy (↑ %) | Tie (↓ %) |
|------|------|------|
| 5 | 81.96 | 3.87 |
| 10 | 82.34 | 3.18 |
| 15 | 82.49 | 1.59 |
| 20 | 82.79 | 1.59 |
| 25 | 83.47 | 1.52 |
| **30** | **84.15** | **1.06** |

Table 3 shows that the selection of the lowest negative log-likelihood candidate (NLL) performs poorly, in some cases worse than Random decoding (e.g., ARC-C: 70.05% vs. 60.84%). In contrast, HEX's frequency-based majority vote achieves much higher accuracy (74.57%), confirming that consensus among diverse trajectories is more reliable than model confidence scores. This shows that the key driver of HEX's success is ensemble agreement.

**Tie break and Latency.** HEX defaults to the smallest block size in tie situations, as Table 4 indicates that jointly considering frequency and log-likelihood does not bring a clear advantage. In addition, we present the wall-time latency of HEX and the baseline inference methods in Table 5.

Table 3: Ablations across datasets. **NLL** selects the candidate with the lowest NLL. HEX's tie issue diminishes as the number of samples increases. Block sizes: [8, 16, 32, 64, 128].

| Method | GSM8K Acc (↑%) | GSM8K Tie (↓%) | MATH Acc (↑%) | MATH Tie (↓%) | ARC-C Acc (↑%) | ARC-C Tie (↓%) | TruthfulQA Acc (↑%) | TruthfulQA Tie (↓%) |
|---|---|---|---|---|---|---|---|---|
| *Baselines* | | | | | | | | |
| Random | 50.87 | – | 16.80 | – | 70.05 | – | 42.40 | – |
| top-$k$ | 22.52 | – | 16.60 | – | 47.87 | – | 29.82 | – |
| top-$k$ margin | 24.72 | – | 16.40 | – | 54.18 | – | 28.36 | – |
| d1 (GRPO) | 79.80 | – | 37.20 | – | 82.68 | – | – | – |
| *Likelihood-based* | | | | | | | | |
| **NLL** | 76.72 | 4.09 | 34.40 | 16.00 | 60.84 | 2.99 | 28.07 | 4.24 |
| *HEX* | | | | | | | | |
| **HEX** | 82.18 | 4.09 | 38.40 | 16.00 | 74.57 | 2.99 | 45.91 | 4.24 |
| **HEX ×5 seeds** | **88.10** | **1.36** | **40.00** | **10.20** | **87.80** | **1.11** | **57.46** | **2.78** |

## 5.5 ADDITIONAL ABLATIONS

A detailed analysis of HEX's tie-breaking rule is provided in Appendix B.1. A comprehensive comparison between individual experts and HEX is provided in Appendix B.3. A comparative analysis under extremely restricted block diversity is provided in Appendix B.4. Finally, an analysis of how HEX's performance varies with different output lengths is provided in Appendix B.5.

## 6 CONCLUSION AND LIMITATION

In this work, we study how diffusion-based language models (dLLMs) generate text. We found that their performance is fundamentally tied to the decoding schedule, the order in which tokens are generated. This is because dLLMs implicitly learn a "set" of semi-autoregressive experts during training. Different schedules activate different experts, and choosing the right one is crucial for getting a high-quality answer. This single insight helps explain common dLLM issues, such as why they sometimes stop generating text too early or fail even when they seem confident.

Based on this insight, we introduced HEX (Hidden semi-autoregressive EXperts), a powerful inference method that requires no extra training. Instead of relying on a single schedule, HEX tries many different schedules at once and lets the experts "vote" on the best final answer. By combining the strengths of the entire hidden team, HEX turns the model's flexibility into a reliable tool for boosting performance. On challenging reasoning benchmarks, HEX doesn't just beat standard methods; it even surpasses models fine-tuned with costly techniques like reinforcement learning (GRPO).

HEX has some limitations. It requires more computation at test time, and we have mainly evaluated it on reasoning tasks. Applying this method to more creative areas like open-ended stories, image generation, or long conversations remains a promising area for future work. Further, we have not established any theoretical understanding of HEX, which is a valid scope of future work.

## ACKNOWLEDGMENTS

This work was partly supported by an IITP grant funded by the Korean Government (MSIT) (No. RS-2020-II201361 , Artificial Intelligence Graduate School Program (Yonsei University)). Bedi is supported by Lockheed Martin and DARPA HR0011262E011.

## ETHICS STATEMENT

Our research aims to reveal the untapped potential of diffusion-based Large Language Model (dLLMs) and to enhance reasoning performance across comprehensive tasks without additional training, through test-time scaling. All datasets used in the evaluation are public and widely known, and to the best of our knowledge, we have thoroughly examined and cited research that is potentially or directly related to our work. We clarify that our use of LLMs was strictly limited to polish writing, such as grammatical correction and fluent expression, not for generating the main content of the research.

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

# A QUALITATIVE RESULTS

## A.1 QUALITATIVE ANALYSIS OF BASELINES VS. HEX

Manolo bought five lollipops and four candies that cost $3.20.
If each lollipop costs $0.40, how much will 10 lollipops and 10 candies cost him?
**Ground Truth: 7**

**Top-K margin:**
First, let's determine the cost of one candy. We know that five lollipops and four candies cost $3.20, and each lollipop costs $0.40.
The total cost of five lollipops is:
$$5 \times 0.40 = 2.00$$

This means that the total cost of the four candies is $1.20. Therefore, the cost of one candy is:
$$\frac{1.20}{4} = 0.00$$

Now, we need to find the cost of 10 lollipops and 10 candies. The cost of 10 candies is:
$$10 \times 0.00 = 0.00$$

The cost of 10 lollipops is:
$$10 \times 0.40 = 4.00$$

Therefore, the total cost for 10 lollipops and 10 candies is $4.00.
</reasoning>

<answer>
\boxed{**4.00**}
</answer><|eot_id|><|endoftext|>

**Random:**
First, we need to determine the cost of one candy. We know that Manolo bought five lollipops and four candies that cost $3.20, and each lollipop costs $0.40. This can be expressed as:

$$5 \times 0.400 + 4C = 3.20$$
$$4C = 2.00$$
$$C = \frac{2.00}{4} = 0.5000$$

Now, we know each lollipop costs $0.40 / 100 = $0.40. Each candy costs $0.4000.

To find the total cost of 10 lollipops and 10 candies:

$$10 \times 0.4000 + 10 \times 0.5000$$
$$10 \times 0.400 + 50000 \times 100 = 5.00$$
</reasoning>

<answer>
\boxed{**5.00**}
</answer><|eot_id|><|endoftext|>

**HEX (Ours):**
First, let's determine the cost of one candy. We know that Manolo bought five lollipops and four candies for a total of $3.20, and each lollipop costs $0.40, so the total cost of five lollipops is $(5 \times 0.40 = 2.00)$.

Therefore, the cost of four candies is $(3.20 - 2.00 = 1.20)$.

Since four candies cost $1.20, the cost of one candy is $(1.20 \div 4 = 0.30)$.

Now, we need to find the cost of 10 lollipops and ten candies. The cost of 10 lollipops is $(10 \times 0.40 = 4.00)$

and the cost of 10 candies is $(10 \times 0.30 = 3.00)$.

So, the total cost is $(4.00 + 3.00 = 7.00)$.
</reasoning>

<answer>
\boxed{**7.00**}
</answer><|eot_id|><|endoftext|>

Figure 7: An instance of generated text responses of different decoding strategies.

## A.2 STRUCTURED PATTERNS INHERENT IN AFTEREOT COLLAPSE

Despite being highly unintuitive and unpredictable, the ordering pattern observed in AfterEoT Collapse exhibits a clear structure: although there is no explicit incentive for the model to unmask padding tokens in a *right-to-left* manner during either training or inference, this regularity consistently emerges throughout the entire unmasking process (see Figure 2). This systematic behavior under collapse suggests that the underlying ordering mechanisms in diffusion LLMs — embedded within both the training objective and the inference procedure — may play a more active and influential role than previously recognized.

## A.3 QUALITATIVE ANALYSIS OF SEMI-AR VS. NON-SEMI-AR

As shown in the right side of Figure 8 in confidence based non-semi-AR decoding, the phenomenon where [AfterEoT] tokens accumulate from the end of the output towards the front indicates that the model is assigning high confidence to [AfterEoT] token throughout the unmasking steps.

The input to the dLLM consists of the number of tokens that make up the prompt and the number of tokens in the desired output sequence, and during training it is subject to limits on the input sequence length for parallel computation. For LLaDA-8B-Instruct (GSAI-ML), this limit is 4,096 tokens. However, in the training of reasoning tasks, most of the output finishes within 256 tokens. In other words, the majority of *ground truth* tokens in the output sequence (more than 93.75%) are

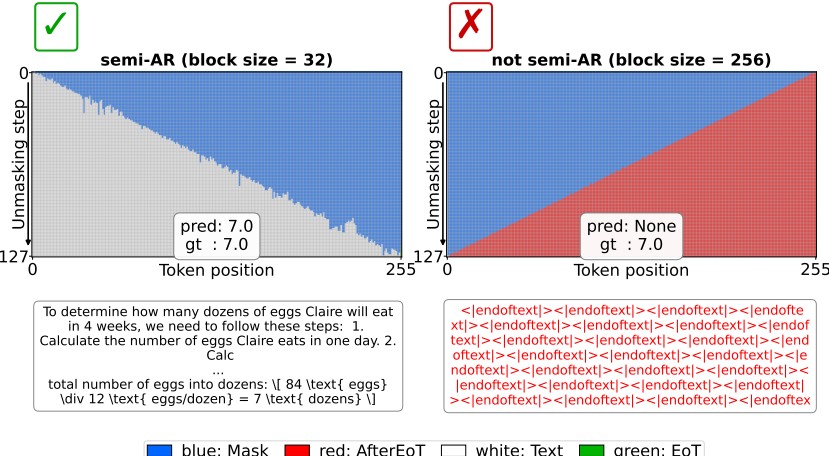

Figure 8: Blue denotes mask tokens, red denotes [AfterEoT] tokens, white denotes text tokens, and green denotes [EoT] tokens (note that in the LLaDA-8-Instruct model, [EoT] and [AfterEoT] are represented as $< |eot\_id| >$ and $< |endoftext| >$, respectively (GSAI-ML)). As unmasking proceeds, two mask tokens are unmasked at each step (output length = 256, unmasking steps = 128). Under a semi-AR regime with block size = 32, positional constraints force reasoning to progress left-to-right while still allowing diffusion-style generation within each block. By contrast, when the positional constraint is removed with block size = 256 (non-semi-AR), the model starts from the last token with the highest confidence—[AfterEoT]—and, due to the inertia of repeatedly generating the same token backward, ultimately collapses into a catastrophic output in which all tokens become [AfterEoT].

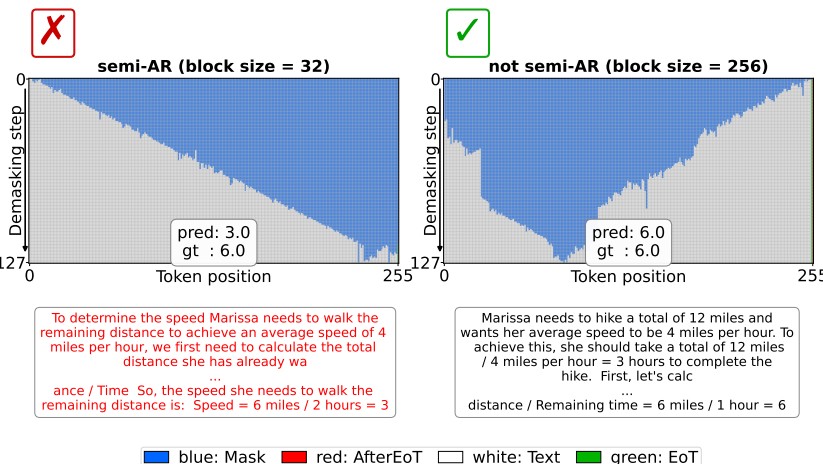

Figure 9: Few cases (2.96%) where the block size of output length hits and semi-AR fails. However, in those cases, the unmasking order converges from both ends toward the center. It means the model commits to an answer before engaging in a full reasoning process—i.e., it states the answer first and only then provides a derivation. This is not only unintuitive; Wang et al. (2025) shows that the answer the dLLM settles on can *flip* repeatedly during the reasoning process. Consequently, when the model locks in an answer first, it is hard to claim that it truly "knows" the answer with certainty.

[AfterEoT]: Given that the training objective is to maximize the average likelihood, we can infer that the dLLM is most strongly taught to generate the [AfterEoT] token.

This suggests that confidence-only decoding is fundamentally limited in its ability to prevent such phenomena during inference, and highlights why the positions of tokens to be unmasked should not be selected based solely on confidence.

Furthermore, confidence-only decoding can lead to issues such as those shown in Figure 9, even when the answer is correct. This arises in reasoning tasks where the correct answer should be derived through step-by-step justification. Instead, the model often generates the answer first without providing supporting reasoning (with the answer tokens generated on the right first), and only afterward begins to generate the reasoning (by unmasking tokens in the middle). Fundamentally, unless the already-generated answer tokens are revised, this generation pattern means that the reasoning cannot actually influence the answer.

Besides, as observed by Wang et al. (2025), the probability distribution corresponding to the answer tokens, whether they are correct or incorrect, continues to undergo substantial changes during the reasoning process. This indicates that the model's confidence in the produced early answer cannot be considered reliable.

## B  ADDITIONAL EXAMPLES AND RESULTS WHICH CAN BE USEFUL

### B.1  EXPERIMENTAL RESULTS OF HEX'S TIE-BREAKING METHODS

Frequency-based majority voting has the characteristic that ties can occur within a finite set of candidates. To determine a single final output, we experimented with various tie-breaking rules as shown in Table 4. We found that likelihood-based selection did not provide any significant benefit. As shown in Figure 10, since we observed that performance tends to degrade as the block size approaches the output length, we adopt selecting the output with the smallest block size in tie situations as our default setting.

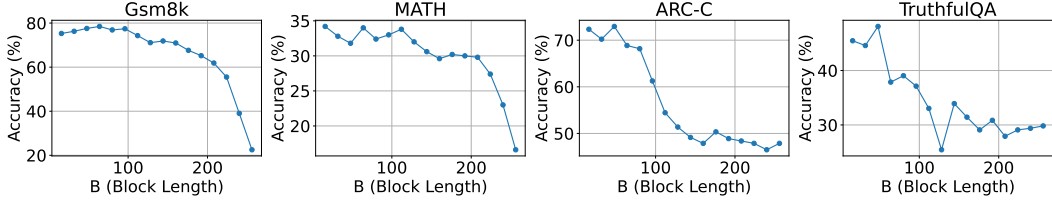

Figure 10: Accuracy across tasks when the block size (B) is increased in fixed increments of 16. Here, block count = (256 // B) when 256 % B is 0 otherwise (256 // B) + 1. Despite the uniform increase in block size, the performance drops drastically in the 128–256 range.

Table 4: Evaluation on tie breaking methods. If the most frequent output is in a tie situation, **TIED: NLL** selects the result with the lowest negative log-likelihood in tie situations, **TIED: first** selects the result generated from the smallest block size when tied, and **TIED: any** treats the case as correct if a correct option exists among the tied candidates. The results of **TIED: any** clearly highlight that majority voting of HEX works well across datasets.

| | GSM8K | | MATH | | ARC-C | | TruthfulQA | |
|---|---|---|---|---|---|---|---|---|
| **Method** | Acc (↑%) | Tie (↓%) | Acc (↑%) | Tie (↓%) | Acc (↑%) | Tie (↓%) | Acc (↑%) | Tie (↓%) |
| *HEX (tie-breaking rules)* | | | | | | | | |
| **HEX, TIED: NLL** | 82.18 | 4.09 | 38.00 | 16.00 | 74.49 | 2.99 | 46.20 | 4.24 |
| **HEX, TIED: first** | 82.18 | 4.09 | 38.40 | 16.00 | 74.57 | 2.99 | 45.91 | 4.24 |
| **HEX, TIED: any** | 83.09 | 4.09 | 41.00 | 16.00 | 76.11 | 2.99 | 47.66 | 4.24 |

### B.2  ANALYSIS OF DECODING LATENCY ACROSS INFERENCE METHODS

For the HEX ×5 seeds setting in Table 3, we further sample 5 independent seeds per expert (5 block sizes × 5 seeds = 25 semi-AR samples). Since each sample has roughly the same cost as a single top-k method, the total token-level inference cost is approximately 25× that of a single top-k run. The additional cost of majority voting over these samples is negligible.

Table 5: Inference efficiency (in seconds) of HEX ($\times$1 seed) on GSM8K, MATH, ARC-C, Truth-fulQA. The numbers in parentheses indicate the number of data points. Random, top-k, and top-k margin use a single sample with a block size of 32. HEX ($\times$1 seed) uses five samples, where each sample is generated with block sizes of 8, 16, 32, 64, and 128. Across all samples, the output length is set to 256, with 2 tokens being unmasked at each step.

| Dataset | Method | Total test set | per batch (8) | per datapoint (1) | ratio |
|---|---|---|---|---|---|
| GSM8K (1319) | random | 2775.73 | 16.76 | 2.09 | $\times$1.0000 |
| | top-k | 2921.70 | 17.64 | 2.20 | $\times$1.0526 |
| | top-k margin | 3187.56 | 19.25 | 2.41 | $\times$1.1484 |
| | HEX | 14613.72 | 88.23 | 11.03 | $\times$5.2648 |
| MATH (500) | random | 1300.62 | 20.46 | 2.56 | $\times$1.0000 |
| | top-k | 1365.98 | 21.51 | 2.69 | $\times$1.0503 |
| | top-k margin | 1477.33 | 23.28 | 2.91 | $\times$1.1359 |
| | HEX | 6823.17 | 107.50 | 13.43 | $\times$5.2461 |
| ARC-C (1172) | random | 2679.99 | 18.15 | 2.27 | $\times$1.0000 |
| | top-k | 2813.69 | 19.05 | 2.38 | $\times$1.0499 |
| | top-k margin | 3048.49 | 20.64 | 2.58 | $\times$1.1375 |
| | HEX | 14062.59 | 95.22 | 11.90 | $\times$5.2473 |
| TruthfulQA (684) | random | 1532.40 | 17.71 | 2.21 | $\times$1.0000 |
| | top-k | 1608.77 | 18.58 | 2.32 | $\times$1.0498 |
| | top-k margin | 1739.78 | 20.12 | 2.52 | $\times$1.1353 |
| | HEX | 8038.26 | 92.90 | 11.61 | $\times$5.2455 |

In practice, these 25 samples can be generated in parallel on modern accelerators, so wall-clock latency can be significantly lower than a strict 25$\times$ factor, but the total inference FLOPs scale linearly with the number of samples. We therefore view the number of experts $\times$ seeds as an explicit accuracy–compute knob: practitioners can trade inference cost for accuracy in a predictable way (Figure 6), without retraining.

## B.3 COMPREHENSIVE COMPARISON WITH THE SEMI-AUTOREGRESSIVE BASELINE

Table 6 illustrates that the average performance of the samples used in HEX's majority voting—that is, the individual semi-AR experts—consistently falls short of HEX across all output lengths (128, 256, and 512). This indicates that dLLMs have implicitly learned a diverse set of inference behaviors from *multiple* semi-AR experts. Therefore, uncovering and leveraging the correct judgment among these hidden experts is essential—an ability that HEX demonstrates with remarkable effectiveness.

| Method | GSM8K | MATH | ARC-C | TruthfulQA |
|---|---|---|---|---|
| 128 [4, 8, 16, 32, 64] $\times$ 1 seed | 68.57 | 28.00 | 79.44 | 48.74 |
| 256 [8, 16, 32, 64, 128] $\times$ 1 seed | 75.39 | 33.16 | 67.27 | 39.59 |
| 512 [16, 32, 64, 128, 256] $\times$ 1 seed | 66.52 | 32.88 | 66.35 | 44.82 |
| 128 [4, 8, 16, 32, 64], **HEX** $\times$ 1 seed | **74.22** | **30.60** | **85.41** | **53.36** |
| 256 [8, 16, 32, 64, 128], **HEX** $\times$ 1 seed | **82.18** | **38.40** | **74.57** | **45.91** |
| 512 [16, 32, 64, 128, 256], **HEX** $\times$ 1 seed | **83.24** | **39.60** | **84.81** | **57.02** |

Table 6: HEX (the bottom three rows) consistently surpasses the mean accuracy (the top three rows) of samples used in majority voting across various output length settings. The number preceding [ ] represents the output length, and the numbers inside [ ] correspond to the block sizes used.

## B.4 COMPARISON BETWEEN HEX AND SAMPLING-BASED VOTING

We compared a single block schedule with 25 different random seeds to HEX (structural diversity: five block schedules $\times$ five seeds). Following common practice, we used a block size of 32 for the single block schedule. HEX with structural diversity outperforms the plain method (with only one block schedule and 25 seeds) by 4.2% in MATH, and 8.57% in GSM8K.

Table 7: HEX outperforms the baseline with a single block schedule and 25 samples.

| Method | MATH (accuracy, tie rate) | GSM8K (accuracy, tie rate) |
|---|---|---|
| semi-AR (single block schedule) | 32.80%, − | 76.27%, − |
| single block schedule (25 seeds) | 35.80%, 4.20% | 79.53%, 0.68% |
| HEX (structural diversity: 5 block schedules × 5 seeds) | **40.00%**, 10.20% | **88.10%**, 1.36% |

**Interpretation.** As shown in Table 7, simply increasing the number of samples for a single block schedule by different seeds does not create sufficient diversity for majority voting; it only perturbs noise, not structure. Structural diversity, using multiple block schedules, changes the underlying context for each semi-AR expert and therefore produces different reasoning trajectories. This is why the plain 25-seeds baseline provides only limited gain, while HEX shows significantly stronger improvements.

## B.5 ABLATION STUDY OF HEX ACROSS DIFFERENT OUTPUT LENGTHS

Across various output lengths, HEX consistently proves to be an effective method. As shown in Table 8, for output lengths of 128, 256, and 512, HEX consistently surpasses both other inference methods and GRPO trained models.

Table 8: Ablation study of HEX across output lengths of 128, 256. The results demonstrate that HEX consistently exhibits robust performance irrespective of output length. Block sizes used are the same as Table 6. Refer to Table 4 for detailed explanation about (b), (c), (d).

| Output Length 128 | | | | |
|---|---|---|---|---|
| Method | GSM8K (acc, tied) | MATH (acc, tied) | ARC-C (acc, tied) | TruthfulQA (acc, tied) |
| Random | 48.82%, − | 20.60%, − | 66.21%, − | 41.96%, − |
| top-$k$ | 53.30%, − | 20.60%, − | 56.74%, − | 36.40%, − |
| top-$k$ margin | 55.57%, − | 22.60%, − | 61.35%, − | 38.30%, − |
| d1 (GRPO) | 72.60%, − | 33.20%, − | 82.68%, − | −, − |
| **NLL (a)** | 70.81%, 7.28% | 29.80%, 20.20% | 78.41%, 1.54% | 47.95%, 4.24% |
| **HEX, TIED: NLL (b)** | 73.69%, 7.28% | 32.00%, 20.20% | 85.24%, 1.54% | 52.63%, 4.24% |
| **HEX, TIED: first (c)** | 74.22%, 7.28% | 30.60%, 20.20% | 85.41%, 1.54% | 53.36%, 4.24% |
| **HEX, TIED: any (d)** | 75.82%, 7.28% | 35.60%, 20.20% | 85.92%, 1.54% | 54.39%, 4.24% |
| **HEX × 5 seeds, TIED: first (e)** | **81.05%, 3.64%** | **33.40%, 15.60%** | **88.40%, 0.60%** | **53.80%, 2.78%** |

| Output Length 256 | | | | |
|---|---|---|---|---|
| Method | GSM8K (acc, tied) | MATH (acc, tied) | ARC-C (acc, tied) | TruthfulQA (acc, tied) |
| Random | 50.87%, − | 16.80%, − | 70.05%, − | 42.40%, − |
| top-$k$ | 22.52%, − | 16.60%, − | 47.87%, − | 29.82%, − |
| top-$k$ margin | 24.72%, − | 16.40%, − | 54.18%, − | 28.36%, − |
| d1 (GRPO) | 79.80%, − | 37.20%, − | 82.68%, − | −, − |
| **NLL (a)** | 76.72%, 4.09% | 34.40%, 16.00% | 60.84%, 2.99% | 28.07%, 4.24% |
| **HEX, TIED: NLL (b)** | 82.18%, 4.09% | 38.00%, 16.00% | 74.49%, 2.99% | 46.20%, 4.24% |
| **HEX, TIED: first (c)** | 82.18%, 4.09% | 38.40%, 16.00% | 74.57%, 2.99% | 45.91%, 4.24% |
| **HEX, TIED: any (d)** | 83.09%, 4.09% | 41.00%, 16.00% | 76.11%, 2.99% | 47.66%, 4.24% |
| **HEX × 5 seeds, TIED: first (e)** | **88.10%, 1.36%** | **40.00%, 10.20%** | **87.80%, 1.11%** | **57.46%, 2.78%** |

| Output Length 512 | | | | |
|---|---|---|---|---|
| Method | GSM8K (acc, tied) | MATH (acc, tied) | ARC-C (acc, tied) | TruthfulQA (acc, tied) |
| Random | 51.71%, − | 16.60%, − | 73.29%, − | 43.86%, − |
| top-$k$ | 9.10%, − | 10.00%, − | 50.17%, − | 29.68%, − |
| top-$k$ margin | 10.08%, − | 9.40%, − | 54.35%, − | 31.14%, − |
| d1 (GRPO) | 81.90%, − | 39.20%, − | 83.28%, − | −, − |
| **NLL (a)** | 78.85%, 4.25% | 25.00%, 20.80% | 63.82%, 5.03% | 34.50%, 7.75% |
| **HEX, TIED: NLL (b)** | 82.11%, 4.25% | 40.20%, 20.80% | 83.11%, 5.03% | 54.97%, 7.75% |
| **HEX, TIED: first (c)** | 82.03%, 4.25% | 40.20%, 20.80% | 84.13%, 5.03% | 55.85%, 7.75% |
| **HEX, TIED: any (d)** | 83.47%, 4.25% | 44.80%, 20.80% | 84.81%, 5.03% | 57.75%, 7.75% |
| **HEX × 5 seeds, TIED: first (e)** | **88.40%, 1.14%** | **47.60%, 10.40%** | **87.12%, 0.85%** | **59.80%, 2.63%** |

## B.6 HOW THE SEMI-AR SCHEDULE LEVERAGES LEARNED PREFIX-LIKE CONTEXTS

Let $x = (x_1, x_2, x_3, x_4)$. To predict $x_4$, the visible set is a subset of $\{1, 2, 3\}$, i.e.

$$U \in \{\varnothing, \{1\}, \{2\}, \{3\}, \{1, 2\}, \{1, 3\}, \{2, 3\}, \{1, 2, 3\}\}.$$

Suppose (due to sequential bias in the data) the model learns well only the prefix-like contexts

$$\widehat{\mathcal{S}} = \{\{1\}, \{1,2\}, \{1,2,3\}\}. \tag{6}$$

Then left-to-right semi-autoregressive (semi-AR) schedules realize exactly these conditionals by changing the block size $B$:

- $B = 1, 1, 1, 1$ (blocks $\{1\}, \{2\}, \{3\}, \{4\}$): when $x_4$ is predicted, the visible set is $U = \{1, 2, 3\}$, so the model uses $p_\theta(x_4 \mid x_1, x_2, x_3)$.

- $B = 2, 2$ (blocks $\{1, 2\}, \{3, 4\}$): when $x_4$ is predicted (with $x_3$ in the same step parellelly), the visible context is the completed first block, $U = \{1, 2\}$, hence $p_\theta(x_4 \mid x_1, x_2)$.

- $B = 3, 1$ (blocks $\{1, 2, 3\}, \{4\}$): when $x_4$ is predicted, $U = \{1, 2, 3\}$ again, hence $p_\theta(x_4 \mid x_1, x_2, x_3)$.

One can also realize $U = \{1\}$, by $B = 1, 3$ (blocks $\{1\}, \{2, 3, 4\}$): when $x_4$ is predicted (with $x_2$, $x_3$ in the same step in parallel), the visible context is the completed first block, $U = \{1\}$, hence $p_\theta(x_4 \mid x_1)$. Furthermore, if we set additional *within*-block order constraints[4] $C_{\text{order}}$ (e.g. descending order of confidence), another possible condition of $U = \{1\}$ can occur by

$$B = 4 \text{ (blocks } \{1, 2, 3, 4\}) \ \wedge \ C_{\text{order}}\big(x_1 \prec x_4 \prec x_2 \prec x_3 \text{ within } B\big),$$

hence the model uses $p_\theta(x_4 \mid x_1)$.

**Key point:** By varying the semi-AR block size and *within*-block order, decoding selects among the learned conditionals in $\widehat{\mathcal{S}}$.

### B.7 ADDITIONAL CONTEXT OF RELATED WORK

**Test-Time Scaling in Autoregressive LLMs.** A growing body of work in autoregressive LLMs has begun to formalize the phenomenon of test-time scaling, the empirical observation that allocating more inference-time compute (e.g., longer reasoning traces or more sampled candidates) can significantly improve model performance without retraining. Muennighoff et al. (2025) introduced budget forcing to emulate the compute-dependent reasoning behaviors, while Aggarwal & Welleck (2025) proposed length-controlled policy optimization to explicitly regulate the reasoning horizon. There are also complementary efforts, such as thinking-optimal scaling (Yang et al., 2025) and adaptive reasoning policies (Arora & Zanette, 2025; Fang et al., 2025; Zhang et al., 2025b; Jiang et al., 2025; Liang et al., 2025; Zhang et al., 2025a; Huang et al., 2025), that highlight the goal of dynamically adjusting cognitive effort based on task complexity and available compute budgets.

In parallel, researchers have explored how controlled sampling and decoding strategies can steer model behavior toward desired objectives without fine-tuning. Best-of-$N$ selection (Stiennon et al., 2020; Nakano et al., 2021; Touvron et al., 2023) provides a simple form of test-time search (which requires a reward signal access) over diverse outputs. We also include an ablation where we compare an instance of Best-of-$N$ with majority vote as the reward signal in Table 9. We note that this is lower than random sampling (cf. Figure 5(a)). On the other hand, more structured methods recast decoding as a reward-guided search (Khanov et al., 2024; Huang et al., 2024; Mudgal et al., 2023; Chakraborty et al., 2024). Recent multi-objective decoding frameworks (Shi et al., 2024) further generalize this idea, balancing several alignment criteria simultaneously.

Table 9: Comparison of majority voting between another non–semi-AR (BoN) and semi-AR experts on GSM8K. The number preceding [ ] represents the output length, and the numbers inside [ ] correspond to the block sizes used.

| Method | Accuracy ($\uparrow$ %) | Tie ($\downarrow$ %) |
|---|---|---|
| 256 [256, 256, 256, 256, 256], BoN with majority voting | 44.35 | 44.58 |
| 256 [8, 16, 32, 64, 128], **HEX** | **82.18** | **4.09** |

---

[4]Adding this condition changes the number of unmasking steps within the block size.

## B.8 Further semi-AR analysis

Figure 11 shows that it is rare for outputs generated with different block sizes under the same prompt to all fail simultaneously. It also reveals that no particular block size consistently dominates in performance. This indicates that heuristically determining the block size for a given prompt can be suboptimal.

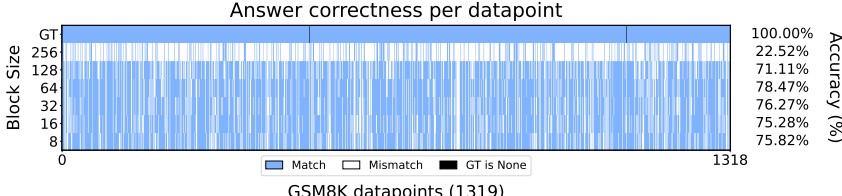

Figure 11: Visualization of the correct (blue) and incorrect (white) answers for each data point in GSM8K when reasoning with various block sizes. Apart from the case where the block size is equal to the output size, i.e., non–semi-AR generation (256), there is no consistent pattern across block sizes in this figure that would allow us to conclude which block size is most suitable for semi-AR reasoning for given tasks. However, the observation that it is rare for all semi-AR outputs to fail on a given data point suggests that the model has the potential to arrive at the correct answer even without additional training, simply by selecting the appropriate decoding policy for the question.

## B.9 Details of dynamic HEX block settings

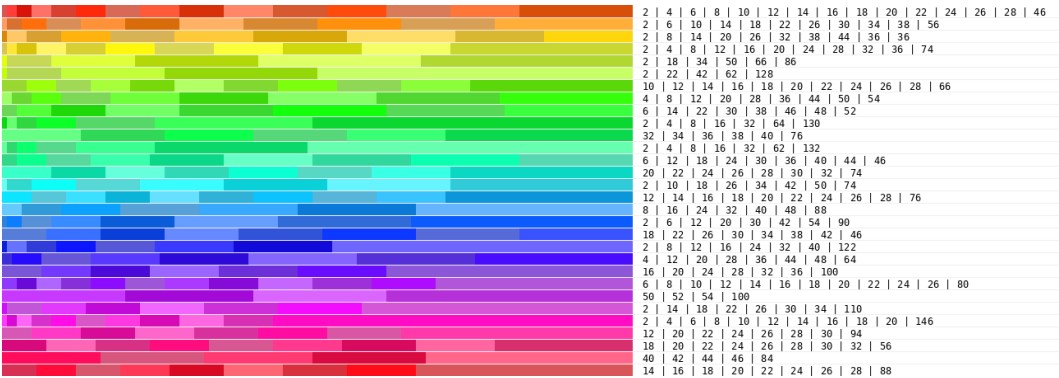

Figure 12: Examples of the block sizes and counts used in the dynamic HEX block settings. Block sizes and counts were randomly chosen and adjusted to match the total output length. The output length is 256 and the number of unmasking steps is 128, meaning that each step unmasks 2 tokens. Accordingly, all block sizes are multiples of 2, and decoding was performed in a semi-autoregressive manner.

