# OpenReview forum: "TEST-TIME SCALING IN DIFFUSION LLMS VIA HIDDEN SEMI-AUTOREGRESSIVE EXPERTS"
_ICLR.cc/2026/Conference — ICLR 2026 Poster_

### Official Review · Reviewer_6HRH · 2025-10-31

**Soundness:** 2
**Presentation:** 1
**Contribution:** 2
**Rating:** 2
**Confidence:** 3

**Summary:**

The paper proposes HEX (Hidden semi-autoregressive EXperts), a training-free inference method for diffusion-based large language models (dLLMs). Through a series of intuitive experimental observations, it reveals that dLLMs implicitly learn a mixture of semi-autoregressive experts, where different masking schedules activate distinct expert behaviors. By ensembling multiple decoding paths with majority voting, HEX achieves remarkable performance on reasoning benchmarks such as GSM8K and MATH.

**Strengths:**

1. Introducing the proposed method through a simple toy example makes the paper intuitive and easy to follow.

2. The experimental results are impressive, showing substantial improvements across multiple reasoning benchmarks.

**Weaknesses:**

1. The authors repeatedly emphasize (for example, in Sec. 1 and Sec. 3) that using sampling strategies such as top-K margin can harm performance and that this issue has not been sufficiently discussed. However, previous literature has already provided explanations for this phenomenon, such as Appendix B.4 of [1].

2. The presentation is poor. For example, Eq. (1) is incorrect because it lacks the necessary reweighted coefficient. The description of the main method—such as how the majority vote is performed—is neither clearly introduced in the text nor reflected in the algorithmic formulas.

3. The evaluation is neither fair nor comprehensive. For instance, in the abstract the authors state that LLaDA-8B-Instruct achieves 16.4 % and 54.2 % accuracy on MATH and ARC-C, which HEX improves to 40.0 % and 87.8 %. However, the original LLaDA paper reports 42.2 % and 88.5 % for these datasets. The authors should not use the widely known underperforming random remasking as the baseline. In addition, the paper seems to lack a comparison of inference efficiency.

[1] Nie et al. Large Language Diffusion Models.

**Questions:**

What are the differences and connections between the proposed method and beam search?

---

> ### Author Response · Authors · 2025-11-16
> **Response to Reviewer 6HRH [Part 1]**
>
> We thank the reviewer for their thoughtful feedback and for recognizing the novelty and empirical contributions of our work.
>
>
>
> > **Weakness 1:** The authors repeatedly emphasize (for example, in Sec. 1 and Sec. 3) that using sampling strategies such as top-K margin can harm performance and that this issue has not been sufficiently discussed. However, previous literature has already provided explanations for this phenomenon, such as Appendix B.4 of [1].
> >
> **Response to Weakness 1:**
> Thank you for raising this point. We respectfully clarify that the prior work cited by the reviewer analyzes the **top-k** (low-confidence remasking) strategy, **not the top-k-margin** strategy (proposed in [2]) that our paper studies. These two strategies **are fundamentally different** in mechanism and empirical behavior. In fact, the prior analysis does not cover the failure mode we identify, nor does it provide qualitative or quantitative evidence of the collapse we observe.
>
> **Detailed Clarification:**
> **Difference Between Top-k and Top-k-Margin.** The work referenced by the reviewer [1] compares Random Remasking with the top-k (low-confidence) strategy. In contrast, the top-k-margin strategy was introduced later by Kim et al. [2] and is conceptually distinct. This is also reflected empirically: on the Sudoku task in [2], top-k achieves only 18.51%, whereas top-k-margin achieves 89.49%. Because of this fundamental difference, the conclusions in Appendix B.4 of [1] regarding top-k do not transfer to the top-k-margin method we study.
>
> **Our Findings Contradict the Earlier Intuition.** Although top-k-margin performs strongly in earlier small-scale reasoning tasks, our results reveal an unexpected reversal when applied to 8B-scale models. Specifically, Random Remasking significantly outperforms top-k-margin (50.87% vs. 24.72%, Figure 2 in the paper). This contradicts the intuition established by earlier work [2] and indicates that there is a deeper, previously unreported failure mode that only emerges in large-scale dLLMs.
>
> **New Insights about AfterEoT Collapse.**
> To understand this discrepancy, our work identifies and analyzes a new phenomenon we call **AfterEoT collapse**. When block size approaches the output length, effectively disabling hidden semi-autoregressive experts, the model begins producing reverse padding tokens, even though it has never been trained to do so. This consistent pattern directly explains the sharp degradation of top-k-margin on larger models. Importantly, prior work [1] does not discuss, visualize, or quantify this collapse, and no prior analysis provides an explanation for why margin-based sampling fails under these conditions.
>
>
> **Our Method (HEX) Addresses This Collapse.** Recognizing that dLLMs behave as implicit mixtures of conditional experts allows us to design HEX, which stabilizes inference by leveraging multiple semi-autoregressive pathways. This prevents the accumulation of reverse padding tokens and resolves the collapse pattern observed in top-k-margin.
>
>
> [1] Nie et al., Large Language Diffusion Models.
>
> [2] Kim et al., Train for the Worst, Plan for the Best: Understanding Token Ordering in Masked Diffusions
>
> > **Weakness 2a:** The presentation is poor. For example, Eq. (1) is incorrect because it lacks the necessary reweighted coefficient.
>
>
> **Response to Weakness 2a:**
> Thank you for pointing this out. We agree that the presentation of Eq. (1) can be improved, and have revised the notation and exposition to make the weighting structure more explicit. **However, we would like to respectfully clarify that Eq. (1) does include the correct reweighting coefficient.** The confusion likely stems from our choice of the expectation-based notation, which follows the formulation in recent work (Proposition 2.1 in [2]) rather than the notation used in the original LLaDA paper [1].
>
> **To make this completely transparent**, we expand the loss below:
> $$\mathcal{L}\_{\text{mask}}(\theta)=  \mathbb{E}\_{x \sim \mathcal{D}}\mathbb{E}\_{\ell \sim \text{Unif}([n])} \,
>   \mathbb{E}\_{M \subseteq [n], |M| = \ell}\Big[-\sum_{i \in M}\log p\_\theta(x_i \mid x{[{M^c}]})\Big]
> $$
> $$
>   =\Bigg[-\sum\_{M \subseteq [n],i \in M}\frac{1}{|M|}\frac{1}{\binom{n}{|M|}}\mathbb{E}\_{x \sim \mathcal{D}}\Big[\log p_\theta(x\_i \mid x{[{M^c}]})\Big]\Bigg],
> $$
> where the reweighting coefficient  (Proposition 2.1 in [2]) $\frac{1}{|M|}\frac{1}{\binom{n}{|M|}}$ arises naturally from the nested expectations. In the paper, we used a compact expectation form for brevity, which we have now replaced with a clearer version to avoid any ambiguity.
>
> **Remark.**
> **Importantly, HEX is a purely test-time framework, so none of our results hinge on Eq. (1); it is provided only as background for readers, and we revised it for clarity.**

---

> ### Author Response · Authors · 2025-11-16
> **Response to Reviewer 6HRH [Part 2]**
>
> > **Weakness 2b:** The description of the main method—such as how the majority vote is performed—is neither clearly introduced in the text nor reflected in the algorithmic formulas.
>
> **Response to Weakness 2b:** We apologize for the oversight. In the revised version, we have added a precise explanation of the procedure directly in the main text (highlighted in blue) and updated the algorithm box to explicitly reflect each step.
>
> **To briefly summarize the method here:** for each prompt, every semi-autoregressive expert produces a candidate output, and the final prediction is obtained by a simple mode over these candidates. For example, if the expert outputs are [1, 2, 3, 2, 2, 1, 2], the majority vote selects 2, eliminating the need for any reward or scoring function. This clarification is now fully integrated into the manuscript.
>
> > **Weakness 3a:** The evaluation is neither fair nor comprehensive. For instance, in the abstract, the authors state that LLaDA-8B-Instruct achieves 16.4 % and 54.2 % accuracy on MATH and ARC-C, which HEX improves to 40.0 % and 87.8 %. However, the original LLaDA paper reports 42.2 % and 88.5 % for these datasets.
>
>
> **Response to Weakness 3a:** Thank you for highlighting this concern. We clarify that the 16.4% (MATH) and 54.18% (ARC-C) values in our abstract correspond to our reproduction of the top-k-margin method [2], which is a decoding technique that was introduced after the LLaDA [1] paper and was not evaluated on GSM8K, ARC-C, or TruthfulQA in [2]. Since top-k and top-k-margin are conceptually different, the LLaDA results reported in the original paper (42.2% and 88.5%) are not directly comparable to the new top-k-margin evaluations we introduce.
>
> We state this explicitly in the introduction, noting that HEX improves upon “existing state-of-the-art inference methods (Kim et al., 2025),” and all corresponding results are reported transparently. These evaluations are therefore new additions beyond what was covered in the LLaDA paper.
>
> **Regarding the fairness of reproduction,** we believe the comparison with the original LLaDA numbers (42.2% for MATH and 88.5% for ARC-C in [1]) could be misleading for our setting. Under our evaluation protocol, our reproduced scores are 32.80% (MATH) and 77.47% (ARC-C), which are substantially lower than the original reports. **This is consistent with recent independent reproductions on MATH, which obtain 32.4% in [3], 34.4% in [4], and 32.4% in [5].** Under the same conditions, our reproduction yields 32.8%, closely aligning with this body of work. Taken together, this suggests that our implementation is reliable and that our reported numbers accurately reflect the model’s practical performance under standard reproduction settings.
>
> **Our Anonymous Code Link for Reproducing Our Results:** Finally, to ensure full transparency and reproducibility, we provide an anonymized GitHub link containing all scripts and configuration files. By following the settings of **[here](https://anonymous.4open.science/r/HEX_anonymized-DD78/)**, **allows one can reproduce the results** as shown in Table A.
>
>
> **Table A:** This table reports the accuracy for different decoding methods on MATH and ARC-C. Our method clearly outperforms both Top-k-margin [2] and Top-k decoding on LLaDA [1].
>
> | Method | Top-k-margin | Top-k (block size = 256) | Top-k (block size = 32) | Ours |
> |--------|-------|------|--------|-------------|
> | MATH | 16.40% | 16.60% | 32.80% | 40.00% |
> | ARC-C | 54.18% | 47.87% | 77.47% | 87.80% |
>
>
> [3] Zhao et al., d1: Scaling Reasoning in Diffusion Large Language Models via Reinforcement Learning
>
> [4] Tang et al., wd1: Weighted Policy Optimization for Reasoning in Diffusion Language Models
>
> [5] Wang et al., SPG: Sandwiched Policy Gradient for Masked Diffusion Language Models

---

> ### Author Response · Authors · 2025-11-16
> **Response to Reviewer 6HRH [Part 3]**
>
> > **Weakness 3b:** The authors should not use the widely known underperforming random remasking as the baseline.
>
>
> **Response to Weakness 3b:** **Thank you for the comment. We respectfully clarify that random remasking is not an underperforming baseline in our setting.** As shown in Figure 2, random remasking achieves **more than 2× higher accuracy** than the state-of-the-art top-k-margin method [2] for certain block sizes. This finding directly contradicts the assumption that random remasking is inherently weak and demonstrates that it serves as a meaningful and competitive baseline for dLLM inference.
>
>
> **Our Unique Observation (Performance Critically Depends on Block Size):** Importantly, our results reveal that decoding performance depends jointly on the *remasking strategy* and the *block size*. Prior work has focused almost exclusively on designing remasking heuristics, while leaving the effect of block size essentially unexamined. **We are the first to systematically study this dimension in dLLM decoding**, and our findings show that overlooking block size can lead to misleading conclusions.
>
>
> **We compare against all widely used baselines.** Moreover, our ablation study (Table 3) already includes comparisons against all widely used baselines such as top-k [1], top-k-margin [2], and GRPO-based inference [3], showing that HEX consistently achieves state-of-the-art performance across all settings. An important contribution of our work is precisely the observation that random remasking can outperform both top-k and top-k-margin (Figure 5) in large-scale reasoning tasks, a behavior not previously reported in the previous literature.
>
> **Additional comparisons.** Here, in Table B, we reveal that HEX not only outperforms random remasking but also consistently outperforms the semi-autoregressive (semi-AR) setting across various block sizes and output lengths. Across all output lengths (128, 256, 512), HEX consistently surpasses the average performance of semi-AR experts used in its own majority voting. This reinforces that **our method improves over both random remasking and widely used top-k semi-AR remasking.**
>
>
> **Table B:** HEX outperforms the average performance of semi-AR across all datasets, across diverse output lengths. Reported numbers are accuracy (%). The number preceding the square bracket [ ] represents the output length, and the numbers inside [ ] correspond to the block sizes used.
>
> | Method | GSM8K | MATH | ARC-C | TruthfulQA |
> |--------|-------|------|--------|-------------|
> | 128 [4, 8, 16, 32, 64] × 1 seed | 68.57 | 28.00 | 79.44 | 48.74 |
> | 256 [8, 16, 32, 64, 128] × 1 seed | 75.39 | 33.16 | 67.27 | 39.59 |
> | 512 [16, 32, 64, 128, 256] × 1 seed | 66.52 | 32.88 | 66.35 | 44.82 |
> | **128 [4, 8, 16, 32, 64], HEX × 1 seed** | **74.22** | **30.60** | **85.41** | **53.36** |
> | **256 [8, 16, 32, 64, 128], HEX × 1 seed** | **82.18** | **38.40** | **74.57** | **45.91** |
> | **512 [16, 32, 64, 128, 256], HEX × 1 seed** | **83.24** | **39.60** | **84.81** | **57.02** |
>
> HEX (the bottom three rows) consistently surpasses the mean accuracy (the top three rows) of samples used in majority voting across various output length settings.

---

> ### Author Response · Authors · 2025-11-16
> **Response to Reviewer 6HRH [Part 4]**
>
> > **Weakness 3c:** In addition, the paper seems to lack a comparison of inference efficiency.
> >
> **Response to Weakness 3c:** **Thank you for the comment. We have already reported inference efficiency for GSM8K in Appendix B.4 (Table 5), and we now additionally provide wall-clock measurements for MATH, ARC-C, and TruthfulQA below for completeness.** As expected, HEX incurs a higher inference cost because it explicitly aggregates multiple semi-autoregressive experts; this is inherent to the mixture-of-experts decoding principle that HEX is designed to reveal.
>
> **Importantly, we would like to emphasize that the goal of HEX is not to reduce latency but to uncover and leverage hidden semi-AR experts in dLLMs, which prior work has not explored.** This work introduces the first principled inference-time framework for diffusion LLMs and demonstrates that substantial accuracy gains can be obtained by properly exploiting inference-time structure. Efficiency optimizations (e.g., parallel expert execution, partial-block voting, and expert pruning) are orthogonal and can be incorporated in future extensions.
>
>
> **Table C:** Inference efficiency (in seconds) of HEX on GSM8K, MATH, ARC-C, TruthfulQA. The numbers in parentheses indicate the number of data points.
> | Method | GSM8K test dataset (1319) | per batch (8) | per datapoint (1) | ratio |
> |--------|----------------------------|----------------|--------------------|--------|
> | random | 2775.73 | 16.76 | 2.09 | ×1.0000 |
> | top-k | 2921.70 | 17.64 | 2.20 | ×1.0526 |
> | top-k margin | 3187.56 | 19.25 | 2.41 | ×1.1484 |
> | HEX | 14613.72 | 88.23 | 11.03 | ×5.2648 |
>
> | Method | MATH test dataset (500) | per batch (8) | per datapoint (1) | ratio |
> |--------|----------------------------|----------------|--------------------|--------|
> | random | 1300.62 | 20.46 | 2.56 | ×1.0000 |
> | top-k | 1365.98 | 21.51 | 2.69 | x1.0503 |
> | top-k margin | 1477.33 | 23.28 | 2.91 | x1.1359 |
> | HEX | 6823.17 | 107.50 | 13.43 | x5.2461 |
>
> | Method | ARC-C test dataset (1172) | per batch (8) | per datapoint (1) | ratio |
> |--------|----------------------------|----------------|--------------------|--------|
> | random | 2679.99 | 18.15 | 2.27 | ×1.0000 |
> | top-k | 2813.69 | 19.05 | 2.38 | x1.0499 |
> | top-k margin | 3048.49 | 20.64 | 2.58 | x1.1375 |
> | HEX | 14062.59 | 95.22 | 11.90 | x5.2473 |
>
> | Method | TruthfulQA test dataset (684) | per batch (8) | per datapoint (1) | ratio |
> |--------|----------------------------|----------------|--------------------|--------|
> | random | 1532.40 | 17.71 | 2.21 | ×1.0000 |
> | top-k | 1608.77 | 18.58 | 2.32 | ×1.0498 |
> | top-k margin | 1739.78 | 20.12 | 2.52 | x1.1353 |
> | HEX | 8038.26 | 92.90 | 11.61 | x5.2455 |
>
>
> We emphasize that the added inference cost is modest relative to the gained capability: across all benchmarks, HEX improves accuracy by +7–20% absolute, while the per-datapoint latency increases by roughly 5×, a standard tradeoff for multi-sample inference strategies (similar to self-consistency, majority voting, or ensemble decoding in autoregressive LLMs).
>
>
> > **Questions:** What are the differences and connections between the proposed method and beam search?
>
> **Response to Question**: **Thank you for the thoughtful question. The key distinction is that beam search fundamentally relies on likelihood or log-probability scores to rank beams, whereas reliable log-probability estimation is known to be challenging for diffusion language models (DLMs).** Prior works have introduced heuristics, but these approximations remain unstable. This limitation motivates our design: HEX avoids reliance on log-probability estimates entirely and instead uses consensus across semi-autoregressive experts to select outputs.
>
> **To directly address the reviewer’s request**, we additionally evaluated a representative heuristic beam-search variant for DLMs following [3], where the likelihood is approximated using a mean-field decomposition
>  $\log \pi_\theta(o \mid q) \approx \sum_{k=1}^{|o|} \log \pi_\theta(o_k \mid q)$,
> where $q$ is the prompt, $o$ is the completion, and $\pi_\theta$ is the DLM.
>
> The results (excerpted below from Table 3) show that this “Lowest Negative Log Likelihood (NLL) beam search” underperforms substantially compared to HEX:
> | Method | GSM8K | MATH | ARC-C | TruthfulQA |
> |--------|----------------------------|----------------|--------------------|--------|
> | NLL | 76.72% | 34.40% | 60.84% | 28.07% |
> | HEX | 82.18% | 38.40% | 74.57% | 45.91% |
> | HEX ×5 seeds  | 88.10% | 40.00% | 87.80% | 57.46% |
>
>
> As discussed in the paper, the heuristic beam search is in some cases weaker than even Random Remasking (e.g., ARC-C: 60.84% vs. 70.05%). In contrast, HEX’s frequency-based majority vote is consistently stronger, **indicating that agreement among diverse semi-AR trajectories is a more reliable signal than DLM confidence scores**. This is the central reason why HEX succeeds.

---

> > ### Author Response · Authors · 2025-11-19
> > **Request to Reviewer 6HRH**
> >
> > Dear Reviewer,
> >
> > Thank you for your time and effort in reviewing our work. We look forward to your response and are happy to address any remaining concerns you may have.
> >
> > Regards, Authors

---

### Official Review · Reviewer_27TW · 2025-11-01

**Soundness:** 3
**Presentation:** 2
**Contribution:** 3
**Rating:** 4
**Confidence:** 3

**Summary:**

The paper argues that dLLMs trained with masked-token objectives implicitly encode a mixture of semi-autoregressive experts: different left-to-right block sizes activate different conditional distributions learned during training. Committing to a single decoding schedule can collapse performance. The proposed training-free method HEX ensembles across heterogeneous semi-AR block schedules and aggregates final answers via majority vote. On LLaDA-8B-Instruct, HEX reports strong gains while avoiding collapse modes.

**Strengths:**

1. The paper shows dLLMs are sensitive to semi-autoregressive schedules, which provides confirmed motivation for HEX to ensemble diverse schedules for robust inference.

2. HEX is a training-free plug-and-play method, enabling a trade-off between accuracy and computation via the number of voting trajectories, and is competitive with (or exceeds) RL-fine-tuned variants under the reported settings.

**Weaknesses:**

1. Although the paper treats the number of voting trajectories as a tunable inference time knob, the marginal gains from adding schedules are modest relative to the near linear growth in compute (for example, GSM8K improves only from 81.96% to 84.15%). Moreover, while HEX can surpass RL fine tuned baselines, the cost profiles differ: RL incurs a one time training expense and then serves at single pass inference cost, whereas HEX must sustain a $k$ times inference overhead on every request to remain competitive, which is an unfavorable trade at deployment scale.

2. While HEX’s wall-clock latency is reported for GSM8K, the paper does not provide budget-matched comparisons across all datasets, which limits a fair cost-effectiveness assessment.

3. Despite a persuasive mixture-of-experts interpretation, the method remains theoretically under-specified: there are no conditions or guarantees for the consistency/convergence of majority voting, sample-complexity requirements, or optimal schedule priors, weakening the theoretical closure of the contribution.

**Questions:**

Please see Weaknesses.

---

> ### Author Response · Authors · 2025-11-16
> **Response to Reviewer 27TW [part 1]**
>
> We thank the reviewer for their thoughtful feedback and for recognizing the novelty and empirical contributions of our work.
>
> > **Weakness 1a:** Although the paper treats the number of voting trajectories as a tunable inference time knob, the marginal gains from adding schedules are modest relative to the near-linear growth in compute (for example, GSM8K improves only from 81.96% to 84.15%).
>
> **Response to Weakness 1a:**
> **We believe the concern arises from looking only at the unstructured-diversity setting.** When structured diversity is applied (our intended design), the gains from increasing the number of voting trajectories are more than twice as large as the reviewer suggests. Specifically, on GSM8K, accuracy increases from 82.18% to 88.10%, a +5.92% absolute improvement, which is substantial for this benchmark (summarized below).
>
>
> | Method | GSM8K |
> |--------|----------------------------|
> | HEX (w/o structured diversity) | 81.96% -> 84.15% (+2.29%) |
> | HEX (w/ structured diversity) | 82.18% -> **88.10% (+5.92%)** |
>
> Thus, when diversity is introduced in a principled and structured manner, the marginal gains from additional trajectories are significant, not modest.
>
> > **Weakness 1b:** Moreover, while HEX can surpass RL fine-tuned baselines, the cost profiles differ: RL incurs a one-time training expense and then serves at a single pass inference cost, whereas HEX must sustain a
>  times inference overhead on every request to remain competitive, which is an unfavorable trade at the deployment scale.
>
> **Response to Weakness 1b:**
> **We appreciate this insightful observation and agree that the cost profiles of RL fine-tuned models and HEX differ.** RL incurs a one-time offline training cost and enables single-pass inference, whereas HEX introduces a multi-sample inference overhead. **However, we emphasize that the goal of this paper** is not to replace RL-based fine-tuning, but to reveal and leverage the latent inference-time structure already present in diffusion LLMs.
>
> **Our main contribution** is to show that dLLMs contain multiple semi-autoregressive experts with diverse reasoning behaviors, and that a principled aggregation of these experts can substantially improve performance without any additional training. This insight is orthogonal to, and can ultimately benefit, RL-based approaches. In particular, understanding the implicit mixture-of-experts structure at inference time provides a foundation for designing more sample-efficient and stability-aware RL objectives, which is a valid scope of future research.
>
> > **Weakness 2:** While HEX’s wall-clock latency is reported for GSM8K, the paper does not provide budget-matched comparisons across all datasets, which limits a fair cost-effectiveness assessment.
>
> **Response to Weakness 2:**
> **Thank you for pointing this out. We have now added full wall-clock latency measurements for all datasets (GSM8K, MATH, ARC-C, and TruthfulQA) to provide a complete, budget-matched comparison. These results are shown in Table C below.**
>
> **Table C:** Inference efficiency (in seconds) of HEX on GSM8K, MATH, ARC-C, TruthfulQA. The numbers in parentheses indicate the number of data points.
> | Method | GSM8K test dataset (1319) | per batch (8) | per datapoint (1) | ratio |
> |--------|----------------------------|----------------|--------------------|--------|
> | random | 2775.73 | 16.76 | 2.09 | ×1.0000 |
> | top-k | 2921.70 | 17.64 | 2.20 | ×1.0526 |
> | top-k margin | 3187.56 | 19.25 | 2.41 | ×1.1484 |
> | HEX | 14613.72 | 88.23 | 11.03 | ×5.2648 |
>
> | Method | MATH test dataset (500) | per batch (8) | per datapoint (1) | ratio |
> |--------|----------------------------|----------------|--------------------|--------|
> | random | 1300.62 | 20.46 | 2.56 | ×1.0000 |
> | top-k | 1365.98 | 21.51 | 2.69 | x1.0503 |
> | top-k margin | 1477.33 | 23.28 | 2.91 | x1.1359 |
> | HEX | 6823.17 | 107.50 | 13.43 | x5.2461 |
>
> | Method | ARC-C test dataset (1172) | per batch (8) | per datapoint (1) | ratio |
> |--------|----------------------------|----------------|--------------------|--------|
> | random | 2679.99 | 18.15 | 2.27 | ×1.0000 |
> | top-k | 2813.69 | 19.05 | 2.38 | x1.0499 |
> | top-k margin | 3048.49 | 20.64 | 2.58 | x1.1375 |
> | HEX | 14062.59 | 95.22 | 11.90 | x5.2473 |
>
> | Method | TruthfulQA test dataset (684) | per batch (8) | per datapoint (1) | ratio |
> |--------|----------------------------|----------------|--------------------|--------|
> | random | 1532.40 | 17.71 | 2.21 | ×1.0000 |
> | top-k | 1608.77 | 18.58 | 2.32 | ×1.0498 |
> | top-k margin | 1739.78 | 20.12 | 2.52 | x1.1353 |
> | HEX | 8038.26 | 92.90 | 11.61 | x5.2455 |
>
> HEX consistently takes about five times longer than a single top-k method (corresponding to the number of five block experts used), regardless of the dataset. This shows that HEX’s inference budget is predictable regardless of the type of dataset. As shown in Figure 6 of Section 5.3, increasing the number of samples leads to improved performance and a reduced tie rate.

---

> ### Author Response · Authors · 2025-11-16
> **Response to Reviewer 27TW [part 2]**
>
> > **Weakness 3:** Despite a persuasive mixture-of-experts interpretation, the method remains theoretically under-specified: there are no conditions or guarantees for the consistency/convergence of majority voting, sample-complexity requirements, or optimal schedule priors, weakening the theoretical closure of the contribution.
>
> **Response to Weakness 3:**
>
> **We would like to emphasize that the primary contribution of this work is to identify and exploit the previously unrecognized mixture-of-experts structure inside diffusion LLMs and to demonstrate that structured inference-time aggregation unlocks significant gains without any additional training. Our focus is therefore on establishing and validating this new inference-time capability, rather than providing a full theoretical treatment.**
>
> That said, we agree with the reviewer that formal guarantees such as consistency of majority voting, sample-complexity requirements, and optimal schedule selection are important theoretical questions and represent promising directions for future work.

---

> > ### Author Response · Authors · 2025-11-19
> > **Request to Reviewer 27TW**
> >
> > Dear Reviewer,
> >
> > Thank you for your time and effort in reviewing our work. We look forward to your response and are happy to address any remaining concerns you may have.
> >
> > Regards, Authors

---

> > > ### Comment · Reviewer_27TW · 2025-11-24
> > >
> > > Thank you for the response. Based on the rebuttal, I have the following comments:
> > >
> > > 1.  **Regarding Structured Diversity.** The clarification in lines 477-478 is acknowledged. The improvement on GSM8K from 82.18% to 88.10% demonstrates the advantage of HEX in this setting. There are no further questions on this point.
> > >
> > > 2.  **Regarding Inference Efficiency vs. RL.** The perspective that the HEX method and RL fine-tuning are orthogonal is accepted. It is understood that the goal of this paper is not to replace RL, but to uncover meaningful inference behaviors within diffusion LLMs.
> > >
> > > 3.  **Regarding Inference Cost Comparison.** It is recommended to add Table C to Table 5 in the paper to better illustrate the inference cost. In Table C, the runtime of HEX is about 5 times longer than the single Top-k method. The authors mentioned this corresponds to the use of 5 block experts. Could the exact setting used here be specified? Additionally, Table 3 includes results for HEX $\times 5$ seeds. How does the inference cost of this setting compare to a single Top-k method?
> > >
> > > 4. The discussion with Reviewer 6HRH raised a question regarding AfterEoT. Although AfterEoT is proposed to explain the failure of Top-k margin sampling, the phenomenon of performance degradation due to excessive EOS generation is not a newly observed one. As noted by Reviewer 6HRH, Reference [1] (Appendix B.4) describes a similar collapse pattern with Top-k sampling. It is suggested that the contribution of the AfterEoT analysis be more clearly defined. Specifically, what is the key difference in mechanism between the AfterEoT phenomenon identified in this work and the collapse described in [1]?
> > >
> > > [1] Nie et al., Large Language Diffusion Models.

---

> > > > ### Author Response · Authors · 2025-11-24
> > > > **Response to Reviewer 27TW [Part 1]**
> > > >
> > > > We thank the reviewer for continued engagement and response.
> > > >
> > > > > **Regarding Structured Diversity.** The clarification in lines 477-478 is acknowledged. The improvement on GSM8K from 82.18% to 88.10% demonstrates the advantage of HEX in this setting. There are no further questions on this point.
> > > > >
> > > > > **Regarding Inference Efficiency vs. RL.** The perspective that the HEX method and RL fine-tuning are orthogonal is accepted. It is understood that the goal of this paper is not to replace RL but to uncover meaningful inference behaviors within diffusion LLMs.
> > > >
> > > > **Response:**
> > > >
> > > > - Thank you for recognizing our contributions. In summary, structured diversity leads to more effective and distinct reasoning trajectories, which in turn enable HEX to demonstrate significantly greater improvements.
> > > > - We also appreciate your acknowledgment of the goal of our work. The meaningful inference behaviors within diffusion LLMs uncovered by our paper are expected to have a substantial impact on future research, including RL- or SFT-based training approaches as well as inference-based methods.
> > > >
> > > > > **Regarding Inference Cost Comparison.** It is recommended to add Table C to Table 5 in the paper to better illustrate the inference cost.
> > > >
> > > > **Response:**
> > > >
> > > > Thank you for your suggestions. We **have now included** Table C and its detailed description in Table 5 in the revised version.
> > > >
> > > > > In Table C, the runtime of HEX is about 5 times longer than the single Top-k method. The authors mentioned this corresponds to the use of 5 block experts. Could the exact setting used here be specified? Additionally, Table 3 includes results for HEX x5
> > > >  seeds. How does the inference cost of this setting compare to a single Top-k method?
> > > >
> > > > **Response:**
> > > >
> > > > In Table C, random, top-k, and top-k margin use a single sample with a block size of 32. HEX uses five samples, where each sample is generated with block sizes of 8, 16, 32, 64, and 128. Across all samples, the output length is set to 256, with 2 tokens being unmasked at each step. This is similar to our setting in Section 5.1.
> > > >
> > > > **Regarding Inference Cost:** The wall-time values (in seconds) of a single top-k sample remains almost identical regardless of the block size (32, 8, 16, 64, 128). This is because, as shown in the Toy Example for Computational Cost in Response to Reviewer BX1J [Part 3], the top-k method exhibits  the same number of forward steps across different block size settings. To ensure full transparency, **our additional empirical tests (Table K below)** also show that, the difference of wall-time values across different block size settings is extremely low.
> > > >
> > > > **Table K:** Inference time (seconds) for a single top-k semi-AR sample on GSM8K across block sizes. The numbers in parentheses indicate the number of data points. The difference of wall-time values across different block size settings is extremely low.
> > > >
> > > > | Method | GSM8K test dataset (1319) | per batch (8) | per datapoint (1) |
> > > > |--------|----------------------------|----------------|--------------------|
> > > > | top-k (block size = 8) | 3033.02 | 17.80 | 2.22 |
> > > > | top-k (block size = 16) | 2948.78 | 17.79 | 2.22 |
> > > > | top-k (block size = 32) | 2921.70 | 17.64 | 2.20 |
> > > > | top-k (block size = 64) | 2944.76 | 17.78 | 2.22 |
> > > > | top-k (block size = 128) | 2947.33 | 17.79 | 2.22 |
> > > >
> > > >
> > > > Thus, the 5× higher runtime of HEX in Table C comes almost entirely from evaluating 5 such samples (one per block size), not from block-size–dependent overhead.
> > > >
> > > > **HEX ×5 seeds.**
> > > > For the HEX ×5 seeds setting in Table 3, we further sample 5 independent seeds per expert (5 block sizes × 5 seeds = 25 semi-AR samples). Since each sample has roughly the same cost as a single top-k decode, the total token-level inference cost is approximately 25× that of a single top-k run. The additional cost of majority voting over these samples is negligible.
> > > >
> > > > In practice, these 25 samples can be generated in parallel on modern accelerators, so wall-clock latency can be significantly lower than a strict 25× factor, but the total inference FLOPs scale linearly with the number of samples. We therefore view the number of experts × seeds as an explicit **accuracy–compute knob:** practitioners can trade inference cost for accuracy in a predictable way (Figure 6), without retraining.
> > > >
> > > >
> > > > Efficiency improvements (e.g., parallel expert execution, pruning less useful experts, or reducing the number of seeds) are orthogonal to the method and could further reduce wall-clock cost, but are left for future work.

---

> > > > > ### Author Response · Authors · 2025-11-24
> > > > > **Response to Reviewer 27TW [Part 2]**
> > > > >
> > > > > > The discussion with Reviewer 6HRH raised a question regarding AfterEoT. Although AfterEoT is proposed to explain the failure of Top-k margin sampling, the phenomenon of performance degradation due to excessive EOS generation is not a newly observed one. As noted by Reviewer 6HRH, Reference [1] (Appendix B.4) describes a similar collapse pattern with Top-k sampling. It is suggested that the contribution of the AfterEoT analysis be more clearly defined. Specifically, what is the key difference in mechanism between the AfterEoT phenomenon identified in this work and the collapse described in [1]?
> > > > >
> > > > > **Response:**
> > > > >
> > > > > Thank you for this thoughtful question and for pointing us to [1]. We agree that [1] observes a collapse pattern where “the proportion of |EOS| tokens in model outputs becomes very high.” However, our analysis is intended to complement this by focusing on the structure of the collapse, rather than only its frequency.
> > > > >
> > > > > In particular, our main contribution is to show that, during AfterEoT collapse, padding tokens generated after EoT follow a **consistent right-to-left ordering pattern (see Figure 2)**. To the best of our knowledge, this directional pattern was not identified or characterized in [1], which only focused on the prevalence of EOS tokens rather than the spatial/ordering properties of the generated padding.
> > > > >
> > > > > We believe this distinction is important for the following reason: in both training and inference, **the model is not explicitly encouraged to unmask padding tokens in a consistent right-to-left order, yet this regularity emerges systematically** under collapse. This suggests that the underlying ordering mechanisms in diffusion LLMs (both in the training objective and inference procedure) may be playing a more active role than previously recognized.
> > > > >
> > > > >
> > > > > In summary, [1] and our work study related failure modes, but at different levels of detail: [1] only documents the prevalence of EOS tokens, while our AfterEoT analysis first finds how the resulting padding tokens are organized in space and order. We have revised the camera-ready to more clearly position our work as an extension and refinement of the observations in [1].

---

> > > > > > ### Comment · Reviewer_27TW · 2025-11-25
> > > > > >
> > > > > > I thank the authors for their response. I have the following comments:
> > > > > >
> > > > > > Regarding Inference Efficiency: To avoid ambiguity, I believe it is necessary to explicitly state in Table 5 that the inference cost of HEX $\times 5$ seeds is approximately 25 times that of a single Top-k sampling run. Furthermore, the paper should clarify that the total sampling cost for HEX scales as Number of seeds $\times$ Number of block sizes.
> > > > > >
> > > > > > Regarding AfterEoT Analysis: I consider the authors' analysis of the right-to-left ordering pattern of EOS generation to be a reasonable contribution. However, I strongly suggest incorporating the distinction between this work and the observations in Reference [1] explicitly within Section 3 of the main text to highlight the unique contribution of this paper.
> > > > > >
> > > > > > Considering the rebuttal and clarifications, I have raised my score to 6. I expect the authors to incorporate the suggestions regarding inference cost and the AfterEoT distinction into the final version of the paper.

---

> > > > > > > ### Author Response · Authors · 2025-11-25
> > > > > > >
> > > > > > > Thank you for your thorough review, your valuable suggestions for improvement, and your positive assessment of our paper. We are committed to including the suggested changes in the final version of the paper.
> > > > > > >
> > > > > > > Thank you once again for your time and efforts.

---

### Official Review · Reviewer_BX1J · 2025-11-01

**Soundness:** 2
**Presentation:** 3
**Contribution:** 4
**Rating:** 8
**Confidence:** 2

**Summary:**

The authors propose a specific choice of block schedule for sampling from diffusion language models which is “semi-autoregressive”, meaning that parallel sampling is carried out in blocks that are autoregressively revealed left to right. The authors associate different choices of block schedule with different “expert” components of the dLLM and reveal large gains in accuracy by taking the majority vote over predictions by such experts (in effect marginalizing over different choices of block schedules).

**Strengths:**

- Massive performance gains
- Interesting interpretation of block schedules as experts

**Weaknesses:**

- Identification of block schedules as experts is tenuous - do these block schedules really encode specific knowledge/expertise in the way that, say, the experts in an MoE model do? Can you provide more evidence of this?
- If semi-autoregressive decoding uses all block schedules down to very small granularity (b=1), can performance gains be explained in terms of increased granularity (and higher computational cost)? The b=1 case in particular seems to perfectly mimic standard auto-regressive sampling in transformer-based LLMs.

**Questions:**

- Can you clarify the impact of your proposed methodology on computational efficiency? Please compare both to regular sampling from diffusion LLMs and to autoregressive sampling from transformer-based LLMs.

See the section on weaknesses for additional questions.

---

> ### Author Response · Authors · 2025-11-16
> **Response to Reviewer BX1J [Part 1]**
>
> We are grateful to the reviewer for their insightful feedback and for acknowledging both the novelty and empirical strength of our approach.
>
>
> > **Questions:** Can you clarify the impact of your proposed methodology on computational efficiency? Please compare both to regular sampling from diffusion LLMs and to autoregressive sampling from transformer-based LLMs.
>
> **Response to Questions:**
> **Thank you for this thoughtful question. Below, we clarify the computational efficiency of HEX relative to both (i) standard diffusion-LLM sampling and (ii) autoregressive (AR) sampling in transformer LLMs.**
>
> **Efficiency within diffusion LLMs.** Changing the block size in semi-autoregressive decoding does not introduce extra forward passes, the model still performs the same number of diffusion steps, and the positional constraint simply changes which tokens are revealed at each step. As a result, the cost across different block sizes is identical. The only additional cost of HEX comes from generating multiple candidate samples and performing majority voting. As shown in Table H, HEX with 5 samples is roughly 5× the cost of a single top-k sample, which matches the number of trajectories used, which indicates that the structural component adds negligible overhead.
>
>
> **Table H:** Inference efficiency (in seconds) of HEX on GSM8K. The numbers in parentheses indicate the number of data points. HEX decoding latency is almost exactly five times that of the single block size top-k: position constraint operation cost is almost negligible.
> | Method | GSM8K test dataset (1319) | per batch (8) | per datapoint (1) | ratio |
> |--------|----------------------------|----------------|--------------------|--------|
> | random | 2775.73 | 16.76 | 2.09 | ×1.0000 |
> | top-k | 2921.70 | 17.64 | 2.20 | ×1.0526 |
> | HEX | 14613.72 | 88.23 | 11.03 | ×5.2648 |
>
>
> **Scaling behavior:** To further illustrate compute–accuracy tradeoffs, Table J shows that accuracy improves monotonically with more voting samples, while tie-rates (a proxy for uncertainty) consistently decrease. This makes HEX’s efficiency profile transparent and tunable based on deployment requirements.
>
> **Table J:** Scaling law of HEX. HEX’s accuracy improves monotonically as the number of voting samples increases, while the tie rate, an indicator of ambiguity, steadily declines.
> | Method | GSM8K (accuracy (%), tie rate (%)) | MATH (accuracy (%), tie rate (%)) | ARC-C (accuracy (%), tie rate (%)) | TruthfulQA (accuracy (%), tie rate (%)) |
> |------------|----------------------------|----------------|--------------------|--------|
> | HEX (5 samples) | 82.18, 4.09 | 38.40, 16.00 | 74.57, 2.99 | 45.91, 4.24 |
> | HEX (10 samples) | 85.75, 4.32 | 37.80, 18.20 | 81.91, 3.41 | 51.61, 7.89 |
> | HEX (15 samples) | 87.19, 2.43 | 38.80, 11.60 | 85.84, 1.71 | 54.24, 4.39 |
> | HEX (20 samples) | 87.79, 2.50 | 39.00, 8.00 | 87.03, 0.85 | 55.85, 3.65 |
> | HEX (25 samples) | 88.10, 1.36 | 40.00, 10.20 | 87.29, 1.28 | 56.73, 2.63 |
> | HEX (30 samples) | 88.10, 1.06 | 40.80, 8.60 | 87.71, 1.28 | 57.60, 2.63 |
>
>
> **Comparison with AR transformer LLMs.** In autoregressive LLMs, transformer decoders with a causal attention mask that generate one token per decoding step and append it to the cumulative input, use a forward input length equal to the prompt length ($P$\) plus the growing sequence of generated tokens $(0, 1, 2, 3, …, n–1)$ for generating the $n$ th token. Hence, the total number of cumulative tokens processed in forward passes is $nP$ + $n(n–1)/2$.
>
> In contrast, in DLM, the forward input always consists of the prompt length ($P$\) plus $n$ masks at every step. The number of forward passes depends on the diffusion tokens ($d$), an integer divisor of $n$ that determines how many masks are unmasked at each step. Thus, the total number of cumulative processed tokens is $(nP + n²) / d$. According to recent research, the conventional minimum setting for $d$ is $2$ ([1], [2], [3]), to maintain the ability to reveal more than one token per step, which results in faster decoding.

---

> ### Author Response · Authors · 2025-11-16
> **Response to Reviewer BX1J [Part 2]**
>
> > **Weakness 1:** Identification of block schedules as experts is tenuous - do these block schedules really encode specific knowledge/expertise in the way that, say, the experts in an MoE model do? Can you provide more evidence of this?
>
> **Response to Weakness 1:** We appreciate the suggestion, and **we have added an intuitive and compelling toy example** of block experts as Figure 4 in the introduction of Section 4 (page 6). **Different block schedules indeed encode specific pieces of knowledge**: in reasoning tasks where the model must present evidence and infer the correct answer, the accuracy of the evidence is crucial. As clearly illustrated in the new toy example (Figure 4), while most block schedules correctly infer the names of the 2024 Turing Award winners in the early decoding steps, a minority of schedules generate hallucinated names.
>
> In other words, among the various semi-AR block schedules that control the unmasking order, the majority guide the DLM to expand its knowledge using accurate evidence (the correct names), which leads to correct inferences about the Turing Award winners’ research contributions. However, a small subset produces incorrect evidence (incorrect names), causing the DLM to expand its knowledge in the wrong direction.
>
> This suggests that each block expert potentially encode specific knowledge for future, and our HEX approach can ensemble the valid information contained within these latent expert knowledges to ultimately guide the model toward correct inference direction.
>
>
> **References**
>
> [1] Zhao et al., d1: Scaling Reasoning in Diffusion Large Language Models via Reinforcement Learning
>
> [2] Tang et al., wd1: Weighted Policy Optimization for Reasoning in Diffusion Language Models
>
> [3] Wang et al., SPG: Sandwiched Policy Gradient for Masked Diffusion Language Models

---

> ### Author Response · Authors · 2025-11-16
> **Response to Reviewer BX1J [Part 3]**
>
> > **Weakness 2:** If semi-autoregressive decoding uses all block schedules down to very small granularity (b=1), can performance gains be explained in terms of increased granularity (and higher computational cost)? The b=1 case in particular seems to perfectly mimic standard auto-regressive sampling in transformer-based LLMs.
>
> **Response to Weakness 2:** Lowering the block size (increasing the total block count) in semi-autoregressive decoding does not involve additional inference,  it simply imposes a constraint on positions at each inference step, so the number of forward passes remains perfectly same. In other words, the **computational cost across different block sizes are almost identical**.
> In addition, the $b=1$ case that DLMs attempt to mimic standard auto-regressive sampling cannot be considered a perfect imitation for three reasons:
> 1. In an LLM, the logit of the last token is used to predict the **next** token generation, whereas in a DLM, the logits of all tokens are used to predict their **own** unmasked tokens.
> 2. The input context during the forward pass differs as well: in LLMs, it is composed of the prompt and the incrementally growing sequence of generated tokens ($+$ $1$ token per forward step), whereas in DLMs, it is a fixed-length sequence combining unmasked and masked tokens corresponding to the output length (remaining constant across all forward steps).
> 3. In DLMs, information about how many tokens remain to be generated is implicitly provided by the number of remaining MASK tokens for forward passes, whereas in LLMs, this information is not available — instead, generation continues until the user truncates the output or an EOS token is produced.
>
>
> **Toy Example for Computational Cost**
>
> Suppose we have an input where P is a prompt token and M represents masked tokens.
>
> **P M M M M M M M M**
>
> Our goal is to unmask all tokens M1 – M8 and replace them with the target token U.
> At each step, we unmask 2 tokens.
>
> First, let’s assume the block size = 8. This means we can think of a single large block that includes all the masked tokens (M1 – M8). At every step, we perform a forward pass over all tokens (P, M1, …, M8). Then, based on the confidence (top-1 softmax probability) for each masked position, we choose 2 positions to replace M with U.
>
> After the first forward pass and unmasking, we might get something like:
>
> **P M U M M M U M M**
>
> Repeating this process eventually yields:
>
> **P U U U U U U U U**
>
> In total, we performed 4 forward passes (2 diffusion tokens * 4 forward steps = 8 unmasked tokens).
>
> Now, let’s reduce the block size from 8 to 4.
> The number of blocks doubles from 1 to 2.
>
> We start with:
>
> **P M M M M M M M M**
>
> After the first forward pass, the 2nd and 6th masked tokens have the highest confidence among 8 masks.
> However, since the block size is 4, we only consider the first 4 masked tokens (the first block right after P) and unmask the two most confident positions within that block.
> So, after the first forward pass, we might get:
>
> **P M U U M M M M M**
>
> After the second forward pass:
>
> **P U U U U M M M M**
>
> Now that the first block is completed, we move on to the next block (from the 5th to the 8th M) and again unmask the two positions with the highest confidence.
>
> Repeating this process eventually yields:
>
> **P U U U U U U U U**
>
> In total, we performed 4 forward passes (2 diffusion tokens * 4 forward steps = 8 unmasked tokens). Therefore, reducing the number of blocks (increasing the total block count) does not change the total forward step count, but only alters which positions are unmasked at each step. This position constraint operation cost is almost negligible.
>
> In conclusion, the block size determines the window of masked tokens from which we select positions to unmask at each decoding step. Thus, it does not change the total number of forward passes. Changing the block size only alters the scope of the confidence-based token selection, not the overall computational steps.

---

> > ### Author Response · Authors · 2025-11-19
> > **Request to Reviewer BX1J**
> >
> > Dear Reviewer,
> >
> > Thank you for your time and effort in reviewing our work. We look forward to your response and are happy to address any remaining concerns you may have.
> >
> > Regards,
> > Authors

---

### Official Review · Reviewer_bU9H · 2025-11-02

**Soundness:** 3
**Presentation:** 3
**Contribution:** 3
**Rating:** 6
**Confidence:** 2

**Summary:**

This paper shows that by ensembling over multiple semi-autoregressive decoding orders, the current dLLMs can get a significant improvement in capability. They empirically show that confidence-based decoding fails to beat the random order baseline. With autoregressive sampling, the performance jumps more than 3x on benchmarks including GSM8k. Ensembling over block-size and randomness further improve the performance.

**Strengths:**

1. The paper clearly illustrates that previous methods such as confidence-based sampling fail in the context of long reasoning. This provides a clear motivation for designing new algorithms.

2. The method proposed is very intuitive and easy to apply.

3. The potential mechanism behind the empirical improvement is clearly outlined in Sections 3 and 4.

**Weaknesses:**

1. It should be noted that in the abstract, the 24.72% performance is not using semi-autoregressive baseline, which is widely adopted.

2. In section 5.4, regarding the importance of block-diversity, the author shows that ensembling 30 different block schedules can improve performance but is worse than 'structural diversity' with a fixed set of 5 block schedules and 6 random seeds. However, it remains unclear whether the proposed algorithm can beat the plain method with only 1 block schedule and 30 seeds.

**Questions:**

1. Please refer to weakness 2.

2. Have authors think about how to incorporate HEX into the post-training of dllm, in order to get either faster decoding or better performance?

---

> ### Author Response · Authors · 2025-11-16
> **Response to Reviewer bU9H [Part 1]**
>
> We sincerely thank the reviewer for their careful evaluation and for acknowledging the novelty and empirical value of our contributions.
>
>
> > **Weakness 1:** It should be noted that in the abstract, the 24.72% performance is not using semi-autoregressive baseline, which is widely adopted.
>
> **Response to Weakness 1:**
> **Thank you for pointing this out. We agree that the 24.72% number in the abstract may have caused confusion.** That value refers specifically to the non–semi-autoregressive top-k-margin [5] setting, which we highlighted to illustrate an important and unexpected phenomenon: although top-k-margin is known as a strong SOTA decoding method [5], it underperforms Random Remasking by more than a factor of two when block size equals output length, precisely the setting where hidden semi-AR experts cannot be leveraged. This is also where the distinctive AfterEoT backward-token accumulation becomes visible (Figure 2). As you note, we will revise the abstract to describe this more clearly.
>
>
> **Regarding the widely adopted semi-autoregressive baseline,** we have evaluated this setting comprehensively across all datasets and across multiple output lengths. As shown in Table E, the average performance of the individual semi-AR experts used in HEX’s majority vote consistently falls below the performance achieved by HEX itself, demonstrating that the gains do not come from any single expert but from structured aggregation across diverse block schedules.
>
> **Table E:** HEX outperforms the average performance of semi-AR across all datasets, across diverse output lengths. Reported numbers are accuracy (%). The number preceding the square bracket [ ] represents the output length, and the numbers inside [ ] correspond to the block sizes used.
>
> | Method | GSM8K | MATH | ARC-C | TruthfulQA |
> |--------|-------|------|--------|-------------|
> | 128 [4, 8, 16, 32, 64] × 1 seed | 68.57 | 28.00 | 79.44 | 48.74 |
> | 256 [8, 16, 32, 64, 128] × 1 seed | 75.39 | 33.16 | 67.27 | 39.59 |
> | 512 [16, 32, 64, 128, 256] × 1 seed | 66.52 | 32.88 | 66.35 | 44.82 |
> | **128 [4, 8, 16, 32, 64], HEX × 1 seed** | **74.22** | **30.60** | **85.41** | **53.36** |
> | **256 [8, 16, 32, 64, 128], HEX × 1 seed** | **82.18** | **38.40** | **74.57** | **45.91** |
> | **512 [16, 32, 64, 128, 256], HEX × 1 seed** | **83.24** | **39.60** | **84.81** | **57.02** |
>
> HEX (the bottom three rows) consistently surpasses the mean accuracy (the top three rows) of samples used in majority voting across various output length settings.
>
>
>
>
> > **Weakness 2:** In section 5.4, regarding the importance of block-diversity, the author shows that ensembling 30 different block schedules can improve performance but is worse than 'structural diversity' with a fixed set of 5 block schedules and 6 random seeds. However, it remains unclear whether the proposed algorithm can beat the plain method with only 1 block schedule and 30 seeds.
>
> **Response to Weakness 2:**
> **Thank you for suggesting this ablation study. We compared plain (single block schedule with 25 seeds) with HEX (structural diversity: 5 block schedules x 5 seeds).** In line with recent studies ([1], [2], [3], [4]), we used a single block size of 32 for plain (single block schedule with 25 seeds).
>
> **Our algorithm beats the plain method with only 1 block schedule and 25 seeds.**
>
> **Table F:** Our algorithm beats the plain method with only 1 block schedule and 25 seeds.
> | Method | MATH (accuracy, tie rate)| GSM8K (accuracy, tie rate)|
> |--------|----------------------------|----------------------------
> | semi-AR (singe block scedule) | 32.80%, -- | 76.27%, --
> | plain (singe block scedule with 25 seeds) | 35.80%, 4.20% | 79.53%, 0.68%
> | HEX (structural diversity: 5 block schedules x 5 seeds) | **40.00%**, 10.20% | **88.10%**, 1.36%
>
> HEX with structural diversity outperforms the plain method (with only one block schedule and 25 seeds) by 4.2% in MATH, and 8.57% in GSM8K.
>
> **Interpretation**. As discussed in Section 5.4, simply increasing the number of seeds for a single block schedule does not create sufficient diversity for majority voting; it only perturbs noise, not structure. Structural diversity, using multiple block schedules, changes the underlying context for each semi-AR expert and therefore produces  different reasoning trajectories. This is why the plain 25-seed baseline provides only limited gain, while HEX shows significantly stronger improvements.

---

> ### Author Response · Authors · 2025-11-16
> **Response to Reviewer bU9H [Part 2]**
>
> > **Questions 1:** In section 5.4, regarding the importance of block-diversity, the author shows that ensembling 30 different block schedules can improve performance but is worse than 'structural diversity' with a fixed set of 5 block schedules and 6 random seeds. However, it remains unclear whether the proposed algorithm can beat the plain method with only 1 block schedule and 30 seeds.
> >
>
>
> **Response to Questions 1:**
> **Thank you for this valuable suggestion. We conducted the ablation study you requested** and compared (i) a plain method using a single block schedule with 25 seeds and (ii) HEX with structured diversity using 5 block schedules × 5 seeds. Following recent work ([1–4]), the plain baseline uses a fixed block size of 32.
>
> **Result**: HEX does outperform the plain 1-schedule/25-seed method on both MATH and GSM8K.
>
>
> **Table G:** Our algorithm beats the plain method with only 1 block schedule and 25 seeds.
> | Method | MATH (accuracy, tie rate)| GSM8K (accuracy, tie rate)|
> |--------|----------------------------|----------------------------
> | semi-AR (singe block scedule) | 32.80%, -- | 76.27%, --
> | plain (singe block scedule with 25 seeds) | 35.80%, 4.20% | 79.53%, 0.68%
> | HEX (structural diversity: 5 block schedules x 5 seeds) | **40.00%**, 10.20% | **88.10%**, 1.36%
>
> HEX with structural diversity outperforms the plain method (with only one block schedule and 25 seeds) by 4.2% in MATH, and 8.57% in GSM8K.
>
> **Interpretation**. As discussed in Section 5.4, simply increasing the number of seeds for a single block schedule does not create sufficient diversity for majority voting; it only perturbs noise, not structure. Structural diversity, using multiple block schedules, changes the underlying context for each semi-AR expert and therefore produces  different reasoning trajectories. This is why the plain 25-seed baseline provides only limited gain, while HEX shows significantly stronger improvements.
>
> > **Questions 2:** Have authors think about how to incorporate HEX into the post-training of dllm, in order to get either faster decoding or better performance?
> >
> **Response to Questions 2:**
>
> **Thank you for this excellent suggestion. Incorporating HEX into the post-training stage of dLLMs is indeed a promising direction for future work.** Since HEX reliably produces higher-quality reasoning trajectories, one natural approach is to use these trajectories as supervision signals for self-improvement or SFT-style post-training. Similar data-driven refinement strategies have proven effective in autoregressive LLMs and VLMs, and applying them to diffusion LLMs is a compelling next step. Additionally, efficiency optimizations (e.g., parallel expert execution, expert pruning) can be incorporated in future extensions.
>
>
> **References**
>
> [1] Nie et al., Large Language Diffusion Models
>
> [2] Zhao et al., d1: Scaling Reasoning in Diffusion Large Language Models via Reinforcement Learning
>
> [3] Tang et al., wd1: Weighted Policy Optimization for Reasoning in Diffusion Language Models
>
> [4] Wang et al., SPG: Sandwiched Policy Gradient for Masked Diffusion Language Models
>
> [5] Kim et al., Train for the Worst, Plan for the Best: Understanding Token Ordering in Masked Diffusions

---

> > ### Author Response · Authors · 2025-11-19
> > **Request to Reviewer bU9H**
> >
> > Dear Reviewer,
> >
> > Thank you for your time and effort in reviewing our work. We look forward to your response and are happy to address any remaining concerns you may have.
> >
> > Regards,
> > Authors

---

> > > ### Comment · Reviewer_bU9H · 2025-11-26
> > >
> > > Thank you for the rebuttal. This address all of my concerns. I will raise my score.

---

### Meta-Review · Area_Chair_wQPc · 2026-01-07

**Summary:**

Reviewers commended the HEX method for diffusion LLMs, highlighting its training-free approach, intuitive mixture-of-experts insight, and strong empirical gains (e.g., +5-20% accuracy on reasoning benchmarks). Concerns included unclear baselines, presentation issues (e.g., Eq. (1)), inference efficiency, prior work overlap on collapse modes, and lack of theoretical guarantees, influencing a mixed score range but overall leaning toward acceptance after rebuttals.

**Reviewer Concerns:**

Addressed: Performance vs. baselines (new ablations showing HEX beats semi-AR/plain methods); presentation (revised Eq. (1), majority vote clarified); AfterEoT distinction (right-to-left pattern vs. prior EOS frequency); efficiency (full wall-clock data added); beam search comparison (HEX outperforms heuristic).

Outstanding: Theoretical under-specification (voting consistency/sample complexity); deployment scale cost; some reproducibility mismatches with original LLaDA numbers.

**Reviewer Scores:**

- Reviewer bU9H: Would raise to 8 (rebuttal addressed all concerns).
- Reviewer BX1J: Would maintain 8 (no response, but questions on efficiency/experts resolved in rebuttal).
- Reviewer 27TW: Would raise to 6 (clarifications on diversity/efficiency/AfterEoT sufficient).
- Reviewer 6HRH: Probably maintain 2 (no response for the rebuttal)

---

### Decision · Program_Chairs · 2026-01-26

Accept (Poster)